# From Tokens to Thoughts: How LLMs and Humans Trade Compression for Meaning

**Chen Shani**
Stanford University
cshani@stanford.edu

**Liron Soffer**
Tel Aviv University
lironso@mail.tau.ac.il

**Dan Jurafsky**
Stanford University
jurafsky@stanford.edu

**Yann LeCun**
New York University; Meta - FAIR
yann.lecun@nyu.edu

**Ravid Shwartz-Ziv**
New York University
rs8020@nyu.edu

## Abstract

Humans organize knowledge into compact conceptual categories that balance compression with semantic richness. Large Language Models (LLMs) exhibit impressive linguistic abilities, but whether they navigate this same compression-meaning trade-off remains unclear. We apply an Information Bottleneck framework to compare human conceptual structure with embeddings from 40+ LLMs using classic categorization benchmarks (Rosch, 1973a; 1975; McCloskey & Glucksberg, 1978). We find that LLMs broadly agree with human category boundaries, yet fall short on fine-grained semantic distinctions. Unlike humans, who maintain "inefficient" representations that preserve contextual nuance, LLMs aggressively compress, achieving more optimal information-theoretic compression at the cost of semantic richness. Surprisingly, encoder models outperform much larger decoder models in agreement with human categories, suggesting that understanding and generation rely on distinct representational mechanisms. Training-dynamics analysis reveals a two-phase trajectory: rapid initial concept formation followed by architectural reorganization, during which semantic processing migrates from deep to mid-network layers as the model discovers increasingly efficient, sparser encodings. These divergent strategies, where LLMs optimize for compression and humans for adaptive utility, reveal fundamental differences between artificial and natural intelligence. This highlights the need for models that preserve the conceptual "inefficiencies" essential for human-like understanding.

## 1 The Enigma of Meaning in Large Language Models

> "The categories defined by constructions in human languages may vary from one language to the next, but they are *mapped onto a common conceptual space*, which represents a common cognitive heritage, indeed the geography of the human mind." *–Croft (2001) p. 139*

Humans excel at organizing knowledge into concepts which are compact categories that achieve remarkable compression while preserving essential meaning (Murphy, 2004). A single word like "bird" compresses information about thousands of species, yet maintains critical semantic properties (can fly, has feathers, lays eggs). This hierarchical organization (robin → bird → animal; Rosch et al. 1976) represents a fundamental cognitive achievement: balancing efficiency with semantic fidelity.

Large Language Models (LLMs) demonstrate striking linguistic capabilities that suggest semantic understanding (Singh et al., 2024; Li et al., 2025). Yet, a critical question remains unanswered: Do LLMs navigate the compression-meaning trade-off similarly to humans, or do they employ fundamentally different representational strategies? This question matters because true understanding, which goes beyond surface-level mimicry, requires representations that balance statistical efficiency with semantic richness (Tversky, 1977; Rosch, 1973b).

To address this question, we apply Rate-Distortion Theory (Shannon, 1948) and Information Bottleneck principles (Tishby et al., 2000) to systematically compare LLM and human conceptual structures. We digitize and release seminal cognitive psychology datasets (Rosch, 1973b; 1975; McCloskey & Glucksberg, 1978), which are foundational studies that shaped our understanding of human categorization but were previously unavailable in a machine-readable form. These benchmarks, comprising 1,049 items in 34 categories with membership and typicality ratings, offer unprecedented empirical grounding to evaluate whether LLMs truly understand concepts as humans. It also offers much better quality data than the current crowdsourcing paradigm by relying on experts to design and execute the human experiments.

Analyzing embeddings from 40+ diverse LLMs against these benchmarks, we uncover a fundamental divergence: LLMs and humans employ different strategies when balancing compression with meaning. While LLMs achieve broad categorical agreement with human judgment, they optimize for aggressive statistical compression at the expense of semantic nuance. Humans maintain "inefficient" representations that preserve rich, multidimensional structure essential for flexible reasoning.

This divergence manifests across three dimensions. First, LLMs capture categorical boundaries but miss fine-grained semantic distinctions like item typicality, central to human understanding. Second, our information-theoretic analysis reveals LLMs achieve mathematically "optimal" compression-distortion trade-offs, while human categories appear suboptimal. Third, encoder models surprisingly outperform decoder models in human alignment despite smaller scales, indicating that understanding and generation may require fundamentally different representational strategies.

Through analysis of OLMo-7B across 57 training checkpoints, we further uncover how these strategies emerge during learning: conceptual structure develops via rapid initial formation followed by architectural reorganization, with semantic processing migrating from deep to mid-network layers as models discover increasingly efficient encodings.

These findings challenge the assumption that statistical optimality equals understanding. The apparent "inefficiency" of human concepts may reflect optimization for adaptive flexibility. Our framework and newly-digitized benchmarks provide essential tools for monitoring this critical balance, guiding development toward AI systems that achieve not just compression, but comprehension.

## 2 RESEARCH QUESTIONS AND SCOPE

Previous work has explored conceptual representations of LLM through multiple lenses: relational knowledge (Shani et al., 2023; Misra et al., 2021), interpretable concept extraction (Hoang-Xuan et al., 2024; Maeda et al., 2024), sparse activation patterns (Li et al., 2025), similarity preserving architecture Doumbouya et al. (2026), and embedded geometry, including hierarchical structures (Park et al., 2025). While insightful, these studies often lack a deep, quantitative comparison of the compression-meaning trade-off using information theory against rich human cognitive benchmarks.

Separately, cognitive science has applied information theory to human concept learning (Imel & Zaslavsky, 2024; Tucker et al., 2025; Zaslavsky et al., 2018; Sorscher et al., 2022). For example, Zaslavsky et al. (2018) developed an Information Bottleneck framework for color naming efficiency, later extended to animal taxonomies (Zaslavsky et al., 2020). Yet these cognitive studies typically proceed without connecting to modern LLMs, and tend to focus on a specific domain. One notable example is Wu et al. (2025), which examined abstraction transfer in humans and LLMs using a behavioral and cognitive modeling level. Our work is different in the sense that it analyzes how information is preserved or distorted inside LLM embedding spaces under controlled clustering transformations.

These two streams, LLM conceptual analysis and cognitive information theory, rarely intersect. We bridge this gap through rigorous comparison of how LLMs and humans navigate the compression-meaning trade-off, grounding our analysis in established cognitive benchmarks. This leads to three research questions:

> **[RQ1]** *To what extent do LLM-emergent concepts align with human-defined categories?*

> **[RQ2]** *Do LLMs exhibit human-like internal structure, particularly item typicality?*

> **[RQ3]** *How do humans and LLMs differ when balancing compression with semantic fidelity?*

Our framework approaches each RQ through a unified lens. [RQ1] examines the categorical alignment, or how information is *compressed* into discrete groups. [RQ2] probes internal structure, which means how semantic *meaning is preserved* within categories. [RQ3] employs our $\mathcal{L}$ objective to evaluate the integrated trade-off. This progression from compression to preservation to their balance mirrors the fundamental challenge both systems face: creating representations that are simultaneously efficient and meaningful.

Figure 1 overviews the data generation and analyses. Human data was collected by asking whether an item $i$ (e.g., chair) is a good example of the category $C$ (furniture). These ratings are aggregated into ranked similarity profiles for each category. Models generate analogous scores using their embeddings. We then compute three metrics: [RQ1] Mutual Information to assess category recoverability, [RQ2] Spearman correlation to measure alignment with human typicality structure, and [RQ3] a rate-distortion objective capturing the trade-off between representation complexity and meaning preservation.

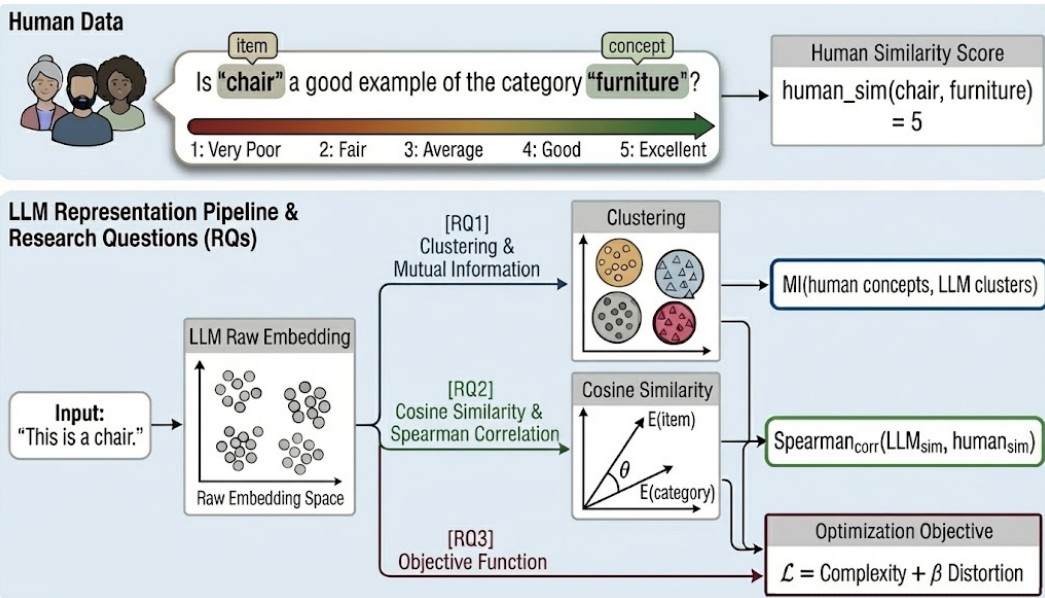

Figure 1: **Overview of the data generation and analyses.** Human data was collected by asking whether an item $i$ (e.g., chair) is a good example of the concept $C$ (furniture). These ratings are aggregated to typicality scores for each category. Models generate analogous scores using their embeddings. We compute three metrics: [RQ1] MI to assess category recoverability, [RQ2] Spearman correlation to measure alignment with human *internal* typicality structure, and [RQ3] a rate-distortion objective capturing the trade-off between representation complexity and meaning preservation.

# 3 BENCHMARKING AGAINST HUMAN COGNITION

Investigating LLM-human conceptual alignment requires robust benchmarks and diverse models. This section details both components.

## 3.1 HUMAN BASELINES: EMPIRICAL DATA FROM SEMINAL COGNITIVE SCIENCE

We draw on three foundational studies that shaped our understanding of human categorization. Unlike many noisy modern crowdsourced datasets, these classic benchmarks were carefully curated by experts, capturing deep cognitive patterns. We focus on three influential works:

**Rosch (1973):** This foundational work (Rosch, 1973a) explored semantic categories as part of the research program leading to prototype theory (Rosch, 1973b)[1]. The theory posits that categories organize around "prototypical" members rather than strict, equally shared features. The dataset includes 48 items in eight common semantic categories (e.g., furniture, bird), with prototypicality rankings (e.g., 'robin' as typical bird, 'bat' as atypical).

**Rosch (1975):** Building on prototype theory, Rosch (1975) further detailed how semantic categories are cognitively represented. This work provides typicality ratings for a larger set of 552 items across ten categories (e.g., 'orange' as a prototypical fruit, 'squash' as less so).

**McCloskey & Glucksberg (1978):** Investigated the "fuzzy" boundaries of natural categories, showing membership is graded rather than absolute (McCloskey & Glucksberg, 1978). Covers 449 items in 18 categories with both typicality scores and membership certainty ratings (e.g., 'dress' is typical clothing, 'bandaid' less so).

While originating from different researchers, these datasets share rigorous experimental designs and provide data on both category assignments and item typicality. We aggregated data from these studies, creating a unified benchmark of 1,049 items across 34 categories. This data, which we have digitized and made publicly available (Appendix C), offers a high-quality empirical foundation for evaluating the human-likeness of LLMs.[2]

## 3.2 LARGE LANGUAGE MODELS UNDER STUDY

We analyze 40+ diverse LLMs spanning multiple architectures and scales (300M to 72B parameters) to understand how conceptual representation varies across model design choices. We note that our analysis requires access to LLMs' embeddings, rather than just output. Thus, we are unable to use any closed-source frontier models such as GPT-5 and Claude.

**Model Selection.** Our study encompasses three architectural paradigms. *Encoder models* include the BERT family (Devlin et al., 2019; He et al., 2021; Zhuang et al., 2021) and CLIP ViT text encoders (Radford et al., 2021). *Decoder models* form the majority of our analysis: the Llama family (1B-70B; Touvron et al., 2023a;b; Grattafiori et al., 2024), Gemma variants (2B-27B; Team et al., 2024; 2025), Qwen models (0.5B-72B; Bai et al., 2023; Yang et al., 2024), Phi series (Javaheripi et al., 2023; Abdin et al., 2024; Abouelenin et al., 2025), Mistral-7B (Jiang et al., 2023), GPT-2 (Radford et al., 2019), and OLMo-7B (**?**). We also include *classic static embeddings* Word2Vec (Mikolov et al., 2013a;b) and GloVe (Pennington et al., 2014) as baselines.

This diverse selection enables us to disentangle effects of architecture (encoder vs. decoder), scale (300M to 72B), and training objectives (understanding vs. generation). Note that encoder-only models are less represented (and are smaller) since recent LLM development has prioritized decoder-only architectures. Complete model specifications appear in Appendix D.

**Embedding Extraction.** We extract representations at two levels to capture different aspects of conceptual knowledge: (1) *static embeddings* from input layers (E matrix), capturing context-free lexical knowledge directly comparable to isolated words in human categorization experiments; and (2) *contextual embeddings* from hidden layers using controlled prompts, revealing how context shapes conceptual structure across network depth.

This dual approach allows us to trace how concepts emerge from basic lexical knowledge to contextualized understanding. Critically, our results prove robust to prompt templates and pooling strategies (Appendix E). Moreover, despite substantial vocabulary overlap between model families, neither token count nor tokenization patterns correlate with our results (Appendix J).

---

[1]Prototype theory is only one account of how humans form concepts; exemplar theory offers an alternative based on stored instances. We do not adjudicate between theories, but use this framework because it provides structured data suitable for modeling. Our computational analysis is compatible with alternative accounts.

[2]Appendix I shows that polysemy is rare in our data and cannot account for our findings.

## 4 A FRAMEWORK FOR COMPARING COMPRESSION AND MEANING

To quantitatively compare how LLMs and humans navigate the fundamental tension between compact representation and semantic richness, we develop a framework that captures both aspects of conceptual organization. Our approach adapts Rate-Distortion Theory (Shannon, 1948) and the Information Bottleneck principle (Tishby et al., 2000) to measure the quality of conceptual systems.

### 4.1 THEORETICAL FOUNDATIONS

Human concepts achieve remarkable efficiency: the word "bird" compresses knowledge about thousands of species into a single category, yet preserves critical semantic information (can fly, has feathers, lays eggs). This reflects a fundamental trade-off that any conceptual system must navigate:

- **Compression:** Grouping diverse items into manageable categories (fewer bits needed)
- **Meaning Preservation:** Maintaining semantic coherence within groups

**Rate-Distortion Theory** (RDT; Shannon, 1948) formalizes this trade-off for lossy compression. Given data $X$ and compressed representation $\hat{X}$, RDT seeks encodings that minimize:

$$R + \lambda D = I(X; \hat{X}) + \lambda \mathbb{E}[d(X, \hat{X})] \tag{1}$$

where $R$ is the rate (bits required), $D$ is distortion (information lost), and $\lambda$ controls their trade-off.

**Information Bottleneck** (IB; Tishby et al., 2000) extends this by compressing $X$ into $Z$ while preserving information about relevant variable $Y$:

$$\min I(X; Z) - \beta I(Z; Y) \tag{2}$$

**Our adaptation:** For conceptual representation, we lack an external relevance variable $Y$. Instead, "relevance" becomes internal semantic coherence, i.e., how well categories preserve within-group similarity. We thus combine RDT's geometric distortion with IB's information-theoretic compression, yielding our framework where clustering $C$ represents items $X$ by minimizing both the information needed to specify items (compression) and the semantic spread within clusters (distortion).

### 4.2 THE $\mathcal{L}$ OBJECTIVE: QUANTIFYING THE TRADE-OFF

We formalize how clustering $C$ represents items $X$ through an objective that combines information-theoretic compression with geometric coherence:

$$\mathcal{L}(X, C; \beta) = \underbrace{I(X; C)}_{\text{Complexity: bits needed}} + \beta \cdot \underbrace{\frac{1}{|X|} \sum_{c \in C} \sum_{e_i \in c} \|e_i - \bar{e}_c\|^2}_{\text{Distortion: semantic spread}} \tag{3}$$

where $\beta$ weights the relative importance of compression versus coherence.

#### 4.2.1 THE COMPLEXITY TERM: MEASURING COMPRESSION

Complexity quantifies how much information the clustering preserves about individual items through mutual information $I(X; C)$. Intuitively, if knowing an item's cluster tells us little about which specific item it is, compression is high (low complexity).

Given $|X|$ items partitioned into clusters of sizes $\{|C_c|\}$:

$$\text{Complexity}(X, C) = I(X; C) = \log_2 |X| - \frac{1}{|X|} \sum_{c \in C} |C_c| \log_2 |C_c| \tag{4}$$

This equals the reduction in uncertainty about item identity when told its cluster. Uniform clusters minimize complexity (one $|X|$ cluster; maximum compression), while singleton clusters maximize it (no compression).

### 4.2.2 THE DISTORTION TERM: MEASURING SEMANTIC COHERENCE

Distortion captures how well clusters preserve semantic relationships and meanings by measuring the average squared distance between items and their cluster centroids in embedding space (spread):

$$\text{Distortion}(X, C) = \frac{1}{|X|} \sum_{c \in C} |C_c| \cdot \sigma_c^2 \tag{5}$$

where $\sigma_c^2 = \frac{1}{|C_c|} \sum_{e_i \in c} \|e_i - \bar{e}_c\|^2$ is the variance within cluster $c$, and $\bar{e}_c$ is its centroid.

Low distortion indicates tight, semantically coherent clusters with similar embeddings. This geometric measure captures the intuition of "meaningful" categories: robins and sparrows cluster tightly as similar birds, while bats would increase distortion on downstream tasks if categorized with them.

### 4.3 CONNECTING FRAMEWORK TO RESEARCH QUESTIONS

Our framework provides unified metrics for all three research questions:

**[RQ1] Categorical Alignment:** How do LLMs and humans partition semantic space? The *Complexity* term $I(X; C)$ directly measures this by quantifies how many bits are needed to specify individual items given their clusters. Comparing $I(X; C_{\text{Human}})$ with $I(X; C_{\text{LLM}})$ reveals whether both systems create similarly-sized groupings with comparable compression rates. Higher mutual information means finer-grained categories; lower means broader, more compressed groupings.

**[RQ2] Internal Semantic Structure:** Do LLMs capture human-like typicality signal? The *Distortion* term measures how well clusters preserve semantic coherence. We tested whether typical items cluster tightly near centroids while atypical items lie farther away. Low distortion with clear center-periphery structure indicates prototype organization that mirrors human cognitive structure.

**[RQ3] Compression-Meaning Trade-off:** How do different systems balance efficiency against semantic fidelity? The complete $\mathcal{L}$ objective reveals fundamental optimization strategies. By varying $K$ (number of clusters) and computing $\mathcal{L}$ curves, we uncover system priorities: aggressive compressors rapidly achieve low $\mathcal{L}$ values by sacrificing nuance, while systems preserving semantic richness maintain higher $\mathcal{L}$ to retain meaningful distinctions. The shape and level of these curves expose whether a system optimizes for statistical efficiency or cognitive utility.

## 5 AN EMPIRICAL INVESTIGATION OF REPRESENTATIONAL STRATEGIES

Building on our information-theoretic framework (Section 4) and established benchmarks (Section 3.1), we empirically investigate how LLMs and humans navigate the compression-meaning trade-off. For each analysis, we examine both static embeddings and contextual embeddings (across all hidden layers), revealing how and when context shapes conceptual organization.

### 5.1 [RQ1] THE BIG PICTURE: ALIGNMENT OF CONCEPTUAL CATEGORIES

We first investigate **whether LLMs form conceptual categories aligned with humans**, which examines how information is *compressed* into discrete groups (the complexity term in our framework).

> **Key Finding: Broad Alignment with Human Categories**
>
> LLM-derived clusters significantly align with human-defined conceptual categories, suggesting they capture key aspects of human conceptual organization. Surprisingly certain encoder models exhibit strong alignment, sometimes outperforming much larger models, highlighting that factors beyond sheer scale influence human-like categorical abstraction.

**Approach:** We tested whether LLMs naturally organize our 1,049 items into categories resembling human conceptual structure. Token embeddings were extracted at two levels: (i) static embeddings from input layers (E matrix), representing context-free lexical knowledge; (ii) contextual embeddings from all hidden layers, measured layer-wise to identify peak conceptual alignment. These were clustered using k-means ($K$ matching human category counts) and evaluated against human

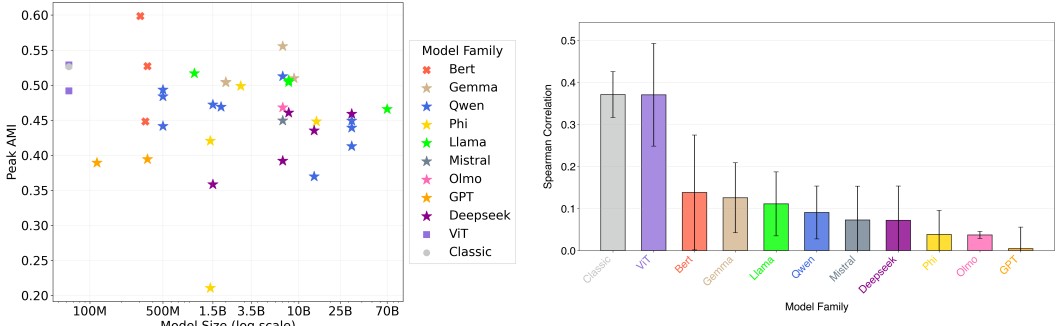

Figure 2: **LLMs capture categorical boundaries (RQ1 AMI scores) but miss internal geometry (RQ2 Spearman correlations). Left:** All 40+ models achieve above-chance AMI with human categories, with encoder architectures (squares, circles, and Xs) matching or exceeding decoder models 100× larger (stars). Results show the layer with peak AMI score per model (see static and mean scores in Figure A.8). **Right:** Despite categorical success, models show weak correlations ($\rho < 0.2$ for most) with human typicality judgments, revealing divergent representational strategies. This divergence between capturing boundaries (compression) while missing internal structure (meaning) reveals how LLMs and humans fundamentally differ in their representational strategies. We note that encoder models align more than decoder models, but these correlations are still modest. Computed using the static embeddings, see full results in Tables 3-4 in Appendix N.

categories using Adjusted Mutual Information (AMI), Normalized Mutual Information (NMI), and Adjusted Rand Index (ARI) metrics. NMI quantifies how much information is shared between the model-derived clusters and the human-labeled categories; AMI refines this measure by correcting for the amount of overlap that would be expected by chance; and ARI assesses the degree of agreement between the two partitions while explicitly accounting for random assignments, providing a complementary view of clustering accuracy.

**Broad Categorical Agreement:** All 40+ models achieve significant above-chance alignment (Figure 2 (Left)). Even at baseline, static embeddings show substantial alignment (mean $AMI \approx 0.45$), which contextual processing enhances to peak $AMI \approx 0.55$. This confirms that LLMs encode human-like categorical boundaries. Full NMI and ARI results in Appendices K-M.

**Architecture Matters More Than Scale:** Surprisingly, BERT-large-uncased (340M parameters) achieves $AMI = 0.60$, matching or exceeding models 100× larger. Classic static Word2Vec and GloVe embeddings, despite predating modern architectures by years, reach AMI scores rivaling contemporary LLMs' peak performance. This suggests that fundamental semantic structure emerges from relatively simple distributional learning, with encoder architectures particularly effective at capturing human-like categories regardless of scale.

## 5.2 [RQ2] ZOOMING IN: FIDELITY TO FINE-GRAINED SEMANTICS

Having established broad categorical alignment, we now examine whether LLMs capture the internal semantic structure of categories. Specifically, we check **how meaning is preserved *within* clusters**.

> **Key Finding: Limited Capture of Semantic Nuance**
>
> While LLMs effectively form broad conceptual categories, their internal representations demonstrate only modest alignment with human-perceived fine-grained semantic distinctions, such as item typicality or psychological distance to category prototypes. This suggests a divergence in how LLMs and humans structure information *within* concepts.

**Approach:** We test whether LLMs encode human-like typicality signals, i.e., whether robins are more "birdy" than penguins. For each item, we compute the cosine similarity with its category name using embeddings (e.g., 'robin' $\rightarrow$ 'bird'; $cosine_{sim}(E(robin), E(bird))$). We compare these similarities with human typicality ratings using Spearman's correlation coefficient $\rho$ (Wissler, 1905).

We employed two analysis approaches: (i) *static-layer analysis* using embeddings directly from the input layer; (ii) *peak AMI layer analysis* using contextual embeddings from the layer that maximized AMI in RQ1. For the peak AMI approach, we extracted category embeddings by replacing items with category names in the same prompt template, ensuring consistent contextualization (see Appendix F for template and pooling robustness analysis).

**Weak Typicality Alignment:** Correlations between LLM internal organization of concepts and human typicality are modest at best (Figure 2 (Right); Tables 3-4 in Appendix N). Static embeddings show weak correlations: BERT achieves $\rho = 0.38$ ($p < 0.05$), while most decoder models fall below $\rho = 0.15$. Even when statistically significant, these correlations indicate limited correspondence with human judgments. This shows that the internal concept geometries of models differ from those of humans, with representation-focused models aligning more closely.

**Architectural Patterns:** Several clear patterns emerge in how different architectures capture typicality. Representation-focused models (Word2Vec, GloVe) and most encoder models (both ViT encoders, BERT-large) demonstrate stronger static-layer performance than decoder-only models (Llama, Gemma, Qwen families). Static correlations range from $\rho \approx 0.25\text{-}0.40$ for representation-focused models versus $\rho < 0.15$ for most decoders.

This divergence likely stems from training objectives: models explicitly trained for representation learning appear more effective at capturing semantic category relationships in their embeddings, while modern decoder-only models, which optimized primarily for next-token prediction, show consistently lower static-layer correlations. The pattern holds across model scales, suggesting architectural design matters more than size for capturing fine-grained semantic similarity.

**Layer-wise Analysis Reveals a Trade-off:** Comparing static and peak AMI layers exposes an architectural limitation. Peak AMI layers, which are optimal for clustering, show systematically weaker typicality correlations than static layers. This pattern holds across model families: layers that best separate categories (RQ1) poorly preserve within-category structure (RQ2). The implication is clear: current architectures encode different aspects of meaning at different depths, forcing applications to choose between broad categorization and semantic nuance.

**Interpretation:** The divergence between LLMs and humans reflects fundamentally different organizational principles. Humans judge typicality through rich, multidimensional criteria: robins are typical birds due to size, flight ability, song, etc. This creates graded categories with clear prototypes and cognitive structures that optimize flexible reasoning and generalization.

LLMs, in contrast, appear to encode flatter statistical associations between items and category labels. Although sufficient for categorization and fluent text generation, these representations miss the prototype structure that makes categories cognitively useful. This difference suggests that LLMs optimize for different objectives than human cognition, a hypothesis that we test directly in RQ3 by examining how each system balances compression against semantic preservation.

## 5.3 [RQ3] THE EFFICIENCY ANGLE: THE COMPRESSION-MEANING TRADE-OFF

Having explored categorical alignment (RQ1) and internal semantic structure (RQ2), we now address our central question: **How do LLM and human representational strategies compare when balancing compression against meaning preservation?**

> **Key Finding: Divergent Efficiency Strategies**
>
> LLMs demonstrate markedly superior information-theoretic efficiency compared to human conceptual structures. Evaluated via our $\mathcal{L}$ objective, LLM-derived clusters consistently achieve more "optimal" compression-meaning balance. Human conceptualizations, while richer, appear less statistically compact, suggesting optimization for cognitive flexibility over pure statistical efficiency.

**Approach:** We analyzed human-defined categories and LLM-derived clusters using our $\mathcal{L}$ objective function (Equation 3, $\beta = 1$) and mean cluster entropy ($S_\alpha$). For LLMs, we performed k-means clustering across various $K$ values to trace the full compression-meaning frontier.

**Results:** Our analysis reveals three key patterns (Figure 3; full results in Appendix Q):

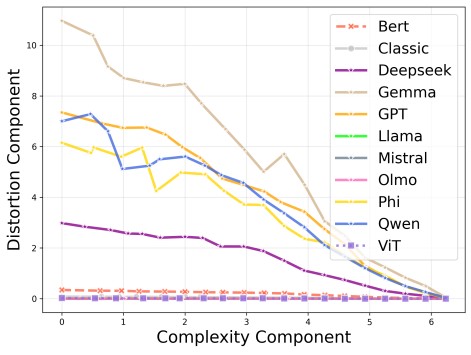
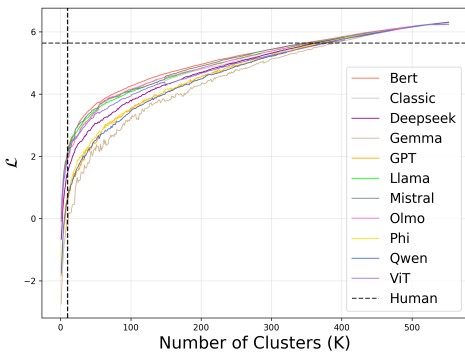

(a) Complexity-distortion trade-offs.

(b) $\mathcal{L}$ comparison with human categories.

Figure 3: **Divergent optimization strategies: LLMs achieve superior information-theoretic efficiency while humans preserve semantic richness. (a)** Encoder models (BERT, ViT, classic static embeddings) consistently achieve lower distortion than decoder models at any given complexity level. **(b)** All LLM-derived clusters achieve lower $\mathcal{L}$ values than human categories (dashed line), indicating more "optimal" compression-distortion balance. Data from Rosch (1975).

*LLMs Achieve Superior Statistical Efficiency:* Both measures reveal stark differences between LLMs and humans (Figure 3; full results in Appendix Q):

*Higher human entropy.* Human concepts consistently exhibit higher cluster entropy than LLM clusters at comparable $K$ values, indicating less statistical compactness but greater internal diversity.

*Lower LLM $\mathcal{L}$ scores.* LLM-derived clusters achieve significantly lower $\mathcal{L}$ values than human categories across all tested $K$ (Figure 3b). Since lower $\mathcal{L}$ signifies more optimal compression-distortion balance, LLMs are demonstrably more "efficient" by this information-theoretic measure.

*Architectural differences.* The Complexity-Distortion plot (Figure 3a) reveals that encoder models (BERT, ViT, classic static models) achieve superior trade-offs. Their distortion at any given complexity is lower compared to decoder models, across both static and contextual embeddings.

**Statistical Optimality Versus Cognitive Utility:** This divergence reveals fundamental differences in optimization pressures. LLMs, trained on massive text corpora, develop maximally efficient statistical representations that minimize redundancy and internal variance. Although human conceptual systems may look suboptimal under information-theoretic measures, prior work indicates that they are structured to support goals such as flexible generalization and causal reasoning rather than maximal compression (Murphy, 2004). Thus, the differing pressures shaping human and LLM representations help explain this apparent suboptimality.

Our analysis reveals that compression efficiency does not predict functional capability. We find no correlation between $\mathcal{L}$ scores and downstream performance ($r = -0.20$, $\rho = 0.51$ on MMLU; Appendix S). This suggests that apparent human "inefficiency" reflects optimization for cognitive flexibility rather than statistical compression. While LLMs excel at compact representation, they may sacrifice the semantic richness essential for human-like understanding.

The consistent architectural patterns observed raise fundamental questions: Do understanding and generation require distinct computational strategies? The superiority of representation-focused models suggests that current architectures may be conflating two fundamentally different cognitive tasks.

## 5.4 EMERGENCE DURING TRAINING: HOW DIVERGENT STRATEGIES DEVELOP

Having established that LLMs and humans employ divergent representational strategies, we investigate *how* these strategies emerge. Analysis of OLMo-7B across 57 training checkpoints (1K to 557K steps, approximately 4B to 2.5T tokens) reveals how conceptual structure emerges.

**Two-Phase Representational Development.** Conceptual organization emerges via two phases (Figure 15). First, rapid concept formation (1K-100K steps) establishes basic categorical structure, with AMI rising from near zero to approximately 0.45, achieving 80% of final alignment within just

10% of training. Second, architectural reorganization (100K-500K steps) systematically migrates semantic processing from deeper layers toward mid-network while AMI continues to gradually improve. This migration from layer 29 to layer 23 occurs without sacrificing categorical alignment, suggesting the model discovers more efficient internal representations. Moreover, this double-phase dynamics occurs when testing attention sparsity, effective rank, and $\mathcal{L}$ values. That is, all of these exhibit the same early rapid shift followed by a slower restructuring phase. This convergence across independent metrics indicates that the model is not merely improving categorical alignment, but reorganizing its internal representations toward increasingly efficient structure. See Appendix H.

**Architectural Reorganization as Optimization.** The upward migration of semantic processing hints that the model discovers increasingly efficient encodings. Early training relies on deep, memorization-heavy representations, but as training progresses, the model shifts to distributed mid-network encoding. This reorganization may explain apparent "emergent" capabilities: they arise not from learning fundamentally new information but from more efficient internal organization of existing knowledge.

**Implications.** These dynamics demonstrate that the compression-oriented strategy observed in fully trained models develops from the earliest stages of training. The rapid initial alignment followed by efficiency-focused reorganization suggests models are inherently biased toward statistical compression rather than semantic richness. Achieving human-like representations may require not just different final objectives but fundamentally different learning dynamics that actively maintain semantic diversity throughout development.

## 6  DISCUSSION AND CONCLUSION

We investigated how LLMs and humans navigate the compression-meaning trade-off in conceptual representation. Using information-theoretic analysis of 40+ models against classic cognitive benchmarks, we reveal fundamental differences in their representational strategies.

**Key Findings.** LLMs achieve broad categorical alignment with humans (AMI $\approx 0.55$), successfully partitioning semantic space into recognizable categories. However, they fail to capture the internal structure that makes these categories cognitively useful, as typicality correlations remain weak ($\rho < 0.2$) across model families. Most strikingly, when evaluated on the compression-meaning trade-off, LLMs consistently achieve lower (better) $\mathcal{L}$ scores than human categories, indicating they optimize for statistical efficiency over semantic richness. This pattern holds across architectures, though encoder models surprisingly outperform decoder models in human alignment despite being orders of magnitude smaller. Training dynamics analysis reveals rapid category formation followed by architectural reorganization that shifts semantic processing from deep to mid-network layers, suggesting efficiency optimization continues throughout training.

**Implications.** These findings challenge the assumption that statistical optimality equals understanding. LLMs excel at their training objective, which is minimizing prediction error, but this drives them toward representations that sacrifice semantic nuances. Encoder models' superior alignment with human representations questions the current paradigm of unified, scaled decoder models, **suggesting that language understanding and generation may require distinct architectures and rely on different processes**. Our framework provides quantitative tools for monitoring the compression-meaning balance in future systems.

**Conclusions.** Our findings reveal an apparent paradox, showing that LLMs are simultaneously better and worse than humans. This occurs because **LLMs and humans employ divergent strategies: statistical compression versus semantic richness**, likely reflecting different optimization pressures. While LLMs process billions of tokens efficiently, humans enable flexible reasoning and generalization. Progress toward human-like AI may require preserving the apparent "inefficiencies" that support cognitive flexibility. We provide theoretical understanding, practical metrics, and high-quality digital benchmarks to develop more human-aligned representations. We encourage the community to utilize the data and metrics for future research towards making AI more human-like.

## 7 ETHICS STATEMENT

Our study relies exclusively on publicly available LLMs and digitized datasets from classic cognitive psychology experiments (Rosch, 1973a; 1975; McCloskey & Glucksberg, 1978). No new human subject data was collected, and all benchmark data we release have been properly attributed and curated to preserve research integrity.

We do not foresee privacy, security, or fairness risks arising from our analyses. Our contribution is methodological and theoretical, focusing on representational trade-offs between humans and LLMs. Nevertheless, we acknowledge that insights into model-human divergences could influence how future systems are designed. We caution that optimizing solely for statistical efficiency without considering semantic richness may exacerbate risks of misinterpretation or oversimplification in socially sensitive applications.

We declare no conflicts of interest or external sponsorship that could bias the reported findings.

## 8 REPRODUCIBILITY STATEMENT

We have taken several steps to ensure reproducibility. All digitized human categorization datasets used in our analyses are publicly released in machine-readable form (Appendix B.1). Detailed model specifications, including architectures, scales, and hyperparameters, are provided in Appendix B.2, and we document embedding extraction procedures, pooling strategies, and prompt templates in Appendix B.3-B.4. Full experimental results, including layer-wise analyses, clustering metrics, and training dynamics across checkpoints, are reported in the appendices (B.5-B.14). Our theoretical framework and derivations are described in Section 4, with complete definitions and formulations provided to enable replication. We will release the code for dataset processing, embedding extraction, and evaluation upon acceptance (to preserve anonymity).

## 9 ACKNOWLEDGMENTS

We are grateful to our teacher and mentor, the late Prof. Naftali Tishby, for his profound contributions to information theory, as well as for inspiring us with the depth, elegance, and excitement of this field. This work was supported in part by the Koret Foundation grant for Smart Cities and Digital Living.

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

# A    COGNITIVE INTUITION

## A.1    A COGNITIVE INTUITION OF THE COMPRESSION-MEANING TRADEOFF

The compression-meaning tradeoff refers to the cognitive tension between representing concepts with maximal efficiency (i.e., minimal information) and preserving the semantic richness needed for flexible generalization, inference, and communication. For instance, upon hearing the sentence "There was a large brown Labrador barking loudly near the playground," a person will often encode a simplified memory, such as "big scary dog near kids." This is not to suggest that humans cannot recall the full sentence, but rather that we typically retain the most meaningful elements to enable efficient reasoning and generalization; without such abstraction and simplification, leveraging past experience for learning and prediction would be more difficult.

We acknowledge that our metrics, while capturing the information-compression trade-off and geometric efficiency in embedding space, do not directly measure all aspects of human-style conceptual abstraction or reasoning. Our aim is not to claim full equivalence with human cognition (nor do we think anyone can or should make such claims), but rather to provide a quantitative, interpretable proxy that highlights where LLMs and humans converge or diverge in how they compress and organize semantic information. We view these metrics as one lens among many, and we are careful in the paper to frame our findings as providing insights into human-like patterns rather than definitive evidence of human-style conceptual processing.

## A.2    COGNITIVELY-INSPIRED INDUCTIVE BIASES

Future models could more closely align with human conceptual structure by incorporating cognitively motivated inductive biases and representational mechanisms. Hierarchical and compositional structure could enable models to capture nested relationships between categories, reflecting the way humans organize knowledge from superordinate to basic-level concepts (e.g., animal → mammal → dog → Labrador). Feature- and relation-based biases could help models focus on meaningful perceptual or functional attributes and relational patterns, rather than relying solely on statistical co-occurrence.

Additionally, theory- or causally grounded priors that draw on humans' intuitive understanding of how objects interact or behave, could constrain learning in complex domains and support more flexible generalization. Incorporating a hybrid exemplar-rule approach, combining memory of specific examples with abstracted rules, would further approximate human category learning. Modular architectures, in which specialized sub-networks handle different aspects of conceptual representation, could enhance generalization and reduce interference between unrelated features. Finally, meta-learned priors distilled from symbolic or program-based representations offer a way to embed structured, human-like concept hypotheses directly into neural models, allowing them to generalize more like humans across novel situations.

Together, these inductive biases offer a path for models that not only compress information efficiently but also organize knowledge in a manner that mirrors human conceptual richness, capturing graded typicality, family resemblance, and hierarchical relationships. While integrating these biases may trade some compression for fidelity, they provide an exciting opportunity to reduce meaning distortion and bridge the gap between statistical efficiency and human-like conceptual understanding.

# B    LIMITATIONS

While this study offers valuable insights, several limitations should be considered.

- Our analysis primarily focuses on English; generalizability across languages with different structures is an open question.
- Human categorization data as a benchmark may not fully capture cognitive complexity and could introduce biases.
- Our IB-RDT objective is applied to specific LLMs; other models or representations might behave differently.
- Our analysis is limited to textual input and does not explore image-based representations.

Future work could address these by expanding to other languages, exploring alternative cognitive models, and testing these principles on different architectures or in real-world applications.

## C    DATASET ACCESS DETAILS

The aggregated and digitized human categorization datasets from Rosch (1973a; 1975); McCloskey & Glucksberg (1978) are made available in CSV format at: [URL deduced for anonymity; Data is attached as Supplementary Material].

## D    LLM DETAILS

- **BERT family:** deberta-large, bert-large-uncased, roberta-large (Devlin et al., 2019; He et al., 2021; Zhuang et al., 2021).
- **QWEN family:** qwen2-0.5b, qwen2.5-0.5b, qwen1.5-0.5b, qwen2.5-1.5b, qwen2-1.5b, qwen1.5-1.5b, qwen1.5-4b, qwen2.5-4b, qwen2-7b, qwen1.5-14b, qwen1.5-32b, qwen1.5-72b, qwen2.5-72b (Bai et al., 2023; Yang et al., 2024).
- **Llama family:** llama-3.2-1b, llama-3.1-8b, llama-3-8b, llama-3-70b, llama-3.1-70b (Touvron et al., 2023a;b; Grattafiori et al., 2024).
- **Phi family:** phi-1.5, phi-1, phi-2, phi-4 (Javaheripi et al., 2023; Abdin et al., 2024; Abouelenin et al., 2025).
- **Gemma family:** gemma-2b, gemma-2-2b, gemma-7b, gemma-2-9b (Team et al., 2024; 2025).
- **Mistral family:** mistral-7b-v0.3 (Jiang et al., 2023).
- **GPT family:** gpt2, gpt2-medium (Radford et al., 2019).
- **DeepSeek family:** DeepSeek-R1-Distill-Qwen-1.5B, DeepSeek-R1-Distill-Qwen-7B, DeepSeek-R1-Distill-Qwen-14B, DeepSeek-R1-Distill-Qwen-32B, DeepSeek-R1-Distill-Llama-8B (DeepSeek-AI, 2025).
- **OLMo family:** Olmo-7b (**?**).
- **ViT family:** Clip ViT-B/32, Clip ViT-B/16 (Radford et al., 2021).
- **Classic static embeddings:** GloVe, Word2Vec (Pennington et al., 2014; Mikolov et al., 2013a;b).

## E    CONTEXTUAL PROMPTS AND POOLING STRATEGIES

Contextual embeddings of LLMs require feeding words into the model through a prompt. Because tokenizers often split a word into multiple tokens, and since some items in our datasets consist of two or more words, we face a design choice regarding how to aggregate token representations. In our methodology, we adopt *average pooling* over the actual tokens, ensuring that all subword pieces contribute equally. Figures 4 and 5 reveal that the average pooling strategy achieves consistent performance and demonstrates the tightest distribution, making it the most reliable choice for our research.

For prompts, we selected a neutral template, `"This is a {word}.  "` (with a trailing space), designed to minimize any additional semantic bias on the target item. Figures 6 and 7 show this prompt to balance performance and consistency, making it ideal for baseline comparisons.

In this section, we explore alternative pooling strategies and evaluate a diverse set of prompt templates across multiple models.

**Pooling Strategies.**    We compare four common approaches:

- **Avg** - mean over all tokens representing the word
- **First** - representation of the first token

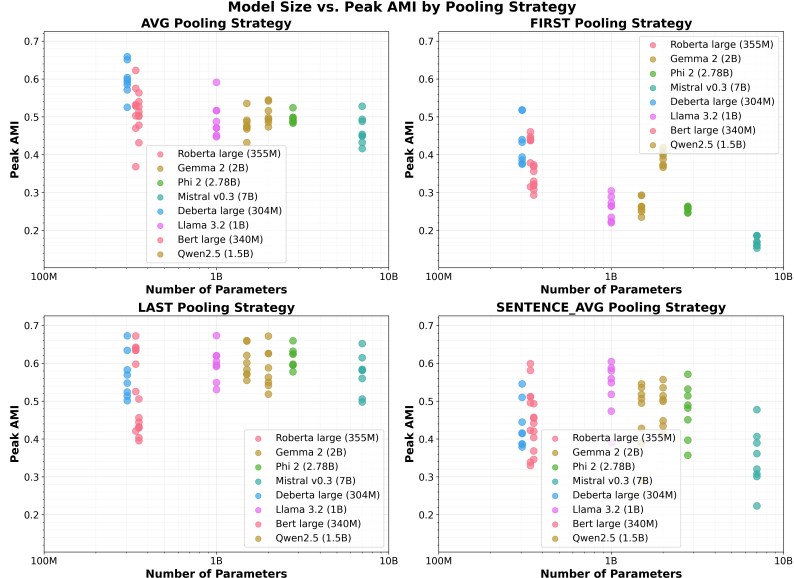

Figure 4: **Average pooling demonstrates consistent performance across different models, making it the most reliable choice for future research.** Each point corresponds to a prompt template applied to a model.

- **Last** - representation of the last token
- **Sentence Avg** - mean over the full sentence embedding

**Prompt Templates.** To test robustness, we design eight templates spanning different linguistic framings:

- `"This is a {word}."`
- `"This is a {word}.    "` (with trailing space)
- `"The concept of {word} is"`
- `"When we think of {word}, we consider"`
- `"A typical {word} would be"`
- `"Examples of {word} include"`
- `"The category {word} contains"`
- `"One kind of {word} is"`

**Models.** We evaluate eight representative LLMs covering major architectures:

- bert-large-uncased (BERT family)
- deberta-large (DeBERTa family)
- gemma-2-2b (Gemma family)
- Llama-3.2-1B (Llama family)
- Mistral-7B-v0.3 (Mistral family)
- phi-2 (Phi family)
- Qwen2.5-1.5B (Qwen family)
- roberta-large (RoBERTa family)

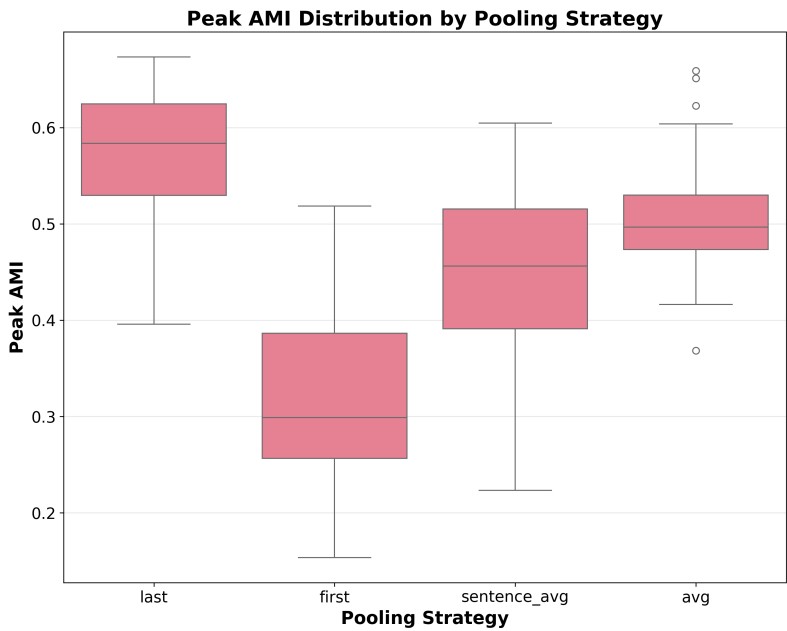

Figure 5: **Average pooling demonstrates the tightest distribution, indicating the highest consistency and reliability across different conditions.** Performance Distribution by Pooling Strategy - Box plots showing AMI distribution for each pooling strategy

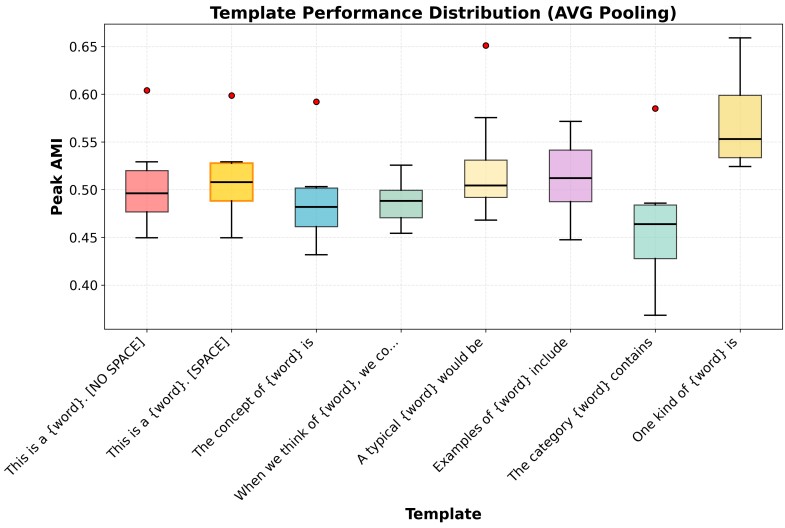

Figure 6: **The neutral prompt template "This is a word. " demonstrates balanced performance and moderate consistency, making it ideal for baseline comparisons.** Performance Distribution across various prompts.

# F  STATIC VS. CONTEXTUAL AMI EXPLORATION

To understand how LLMs develop conceptual alignment with human categories, we examine the progression from static to contextual embeddings. Figure 8 presents three complementary views of this progression across different model scales and architectures.

The left subplot shows **Static AMI** scores, which represent the conceptual alignment achieved by models' input embeddings before any contextual processing (i.e., the E matrix embeddings of the target word). These scores reveal that even at the most basic level, LLMs encode semantic infor-

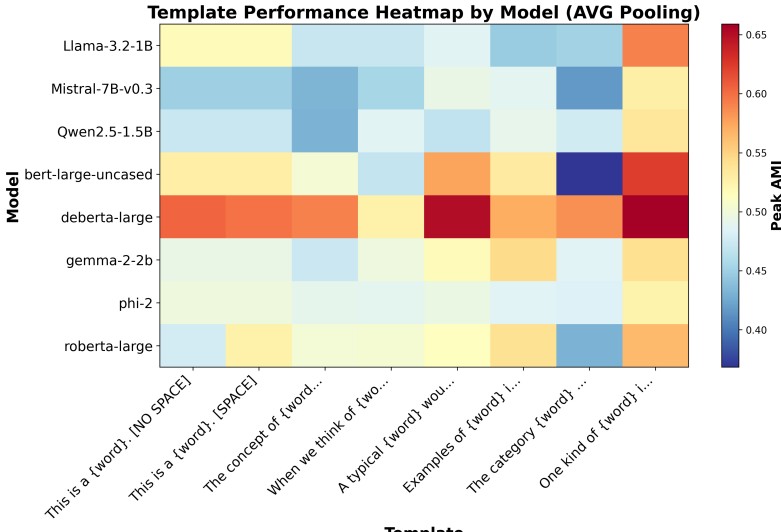

Figure 7: **The neutral template "This is a {word}.   " remains a stable choice across families.** Heatmap showing template performance per model.

mation that supports human-like categorical grouping. Remarkably, static models like Word2Vec and GloVe achieve static AMI scores that rival the peak contextual performance of modern LLMs, suggesting that fundamental conceptual structure is captured early in the learning process.

The middle subplot displays **Average AMI** across all layers, providing a measure of overall semantic representation quality throughout the network. This metric shows the typical performance a model achieves across its entire depth, offering insight into how consistently different layers maintain conceptual alignment. The improvement from static to average AMI demonstrates that contextual processing generally enhances rather than diminishes semantic understanding.

The right subplot reveals **Peak AMI**, representing the optimal conceptual alignment achieved by any single layer. This metric identifies where in the network conceptual understanding is maximized, typically occurring in middle-to-late layers before declining in the final layers. The progression from static to peak AMI shows that contextual processing not only preserves but significantly enhances the conceptual alignment present in static embeddings.

Several key insights emerge from this multi-metric analysis. First, all models demonstrate above-chance alignment even in their static embeddings, confirming that basic semantic structure is a fundamental property of learned representations. Second, the consistent improvement from static to peak AMI across all model types suggests that contextual processing universally enhances conceptual understanding rather than creating it de novo. Third, encoder architectures of different types (BERT, ViT encoders, and static models) achieve comparable or superior performance to much larger decoder models, highlighting that architectural factors and pre-training objectives significantly influence conceptual alignment quality beyond mere model scale.

This analysis complements the main text findings by showing that LLMs do not simply achieve above-chance alignment with human categories but rather so through a systematic progression from basic to sophisticated conceptual representations, with contextual processing serving as an amplifier rather than a generator of semantic understanding.

We also tested the robustness of our results to different clustering seeds (Figure 9). We found AMI to be highly stable across seeds, with negligible variation in the peak values and layer-wise profiles, indicating that our conclusions are not sensitive to the choice of clustering initialization.

Lastly, we plot the peak AMI against the number of FLOPS per token and find no systematic correlation, suggesting that computational cost alone does not predict human-aligned conceptual representations (Figure 10.)

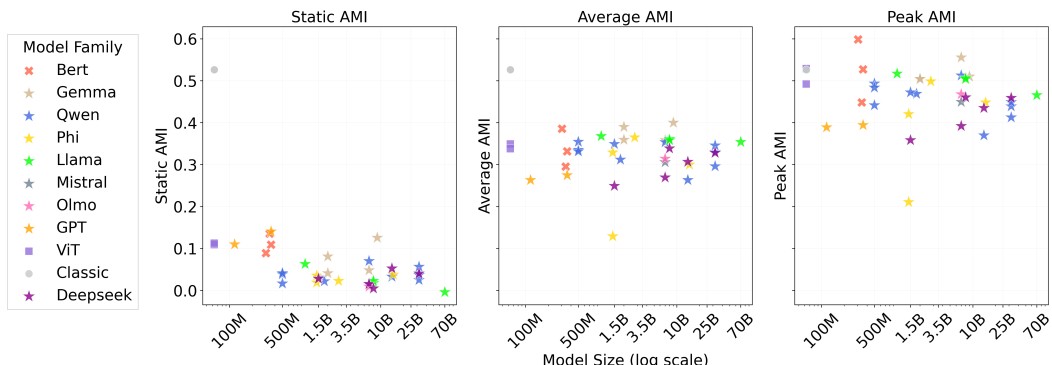

Figure 8: **LLMs begin with basic conceptual alignment in static embeddings and achieve progressively stronger alignment through contextual processing.** Three subplots showing model size (log scale) versus different AMI metrics. **Left:** Static AMI reveals baseline categorical structure. **Middle:** Average AMI across layers shows overall semantic quality. **Right:** Peak AMI demonstrates high conceptual alignment. The consistent improvement from static to peak AMI across all model types reveals that contextual processing enhances rather than creates conceptual understanding, with encoders (BERT, ViT encoders, and static models) achieving comparable or superior performance to much larger decoder models.

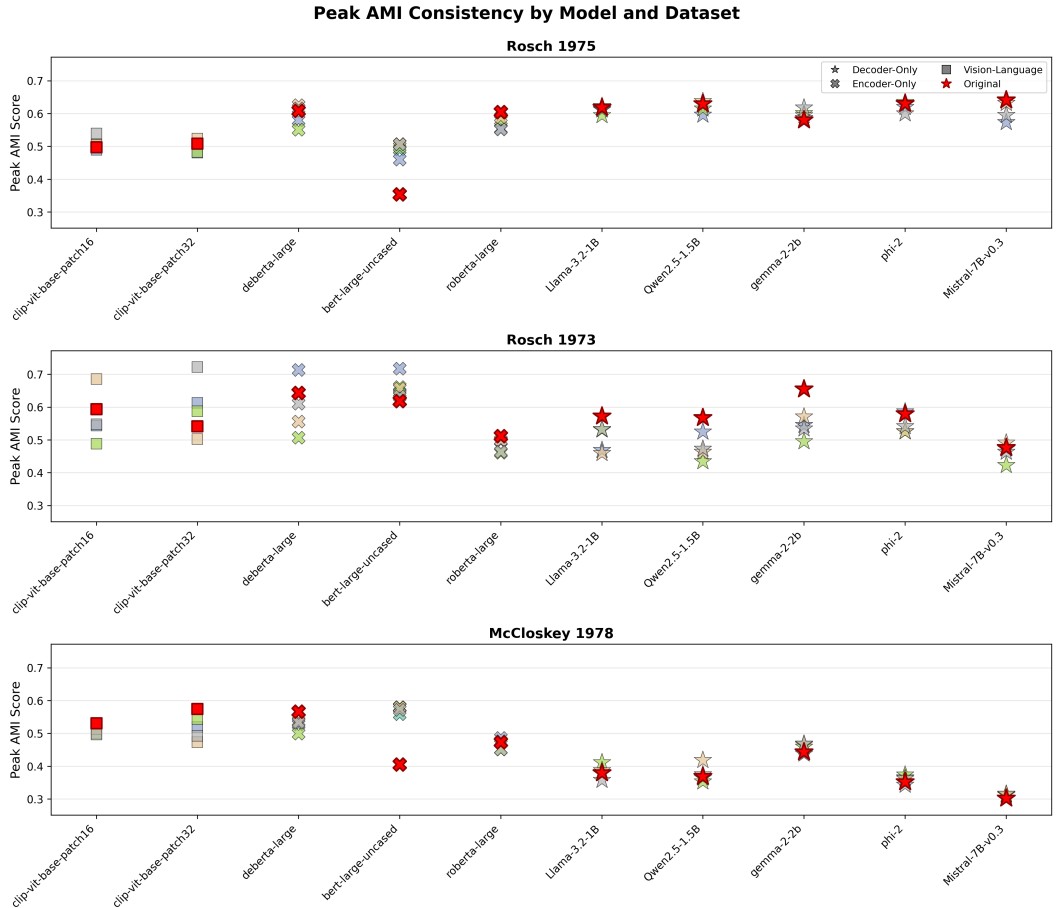

Figure 9: **AMI Scores are Robust Across Clustering Seeds.** Peak AMI between model representations and human categories remains highly stable across multiple random clustering initializations. Peaks in each plot are consistent, indicating that the observed alignment patterns are not sensitive to the choice of clustering seed.

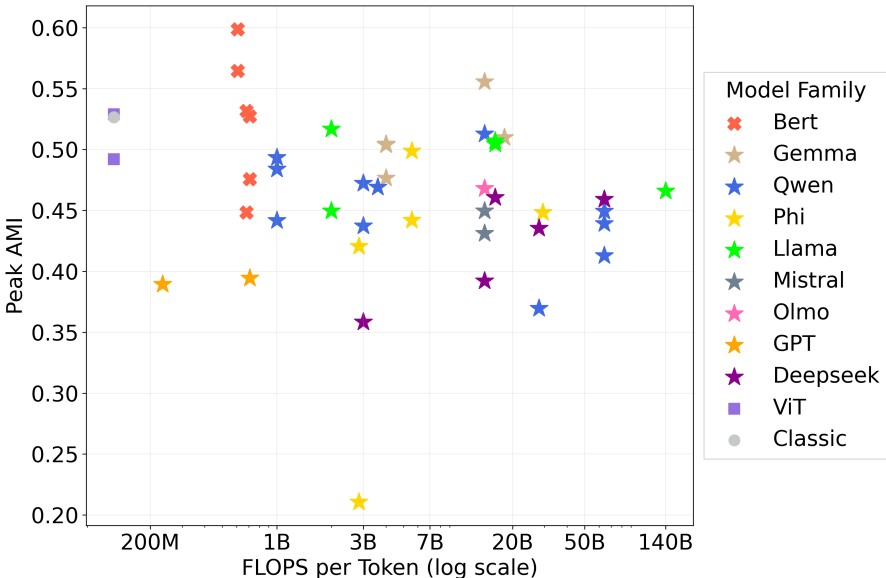

Figure 10: **Peak AMI versus computational cost.** We plot each model's peak AMI against its FLOPS per token. There is no systematic correlation, indicating that higher computational cost does not necessarily lead to better alignment with human conceptual structure.

## G    MULTILINGUAL ANALYSIS

To further explore conceptual understanding across languages, we translated our dataset into Spanish, German, Italian, and Russian using Google Translate's API and repeated our analyses with the same LLMs and methods. In RQ1, a clear scale effect emerges for all non-English languages, while English shows no such trend (Figures 11, 12). We interpret this as a consequence of limited non-English training data: larger models are more likely to have been exposed to sufficient multilingual data, improving their conceptual alignment. In RQ2, all models struggle to preserve the internal geometry of human concepts across languages (Figure 13). RQ3 shows that non-English languages exhibit greater compression (Figure 14), consistent with our explanation for RQ1: smaller exposure to non-English data leads to more compressed representations, reducing flexibility and interpretability.

## H    TRAINING DYNAMICS

The OLMo analysis examines how semantic structure develops during training by analyzing 57 intermediate checkpoints from the OLMo-7B model, representing evenly spaced sampling (every 10K training steps) spanning from 1K to 557K steps (covering approximately 4B to 2.5T tokens).

The analysis employs two complementary sampling strategies: **representative sampling (6 checkpoints)** captures major developmental phases at 1K, 101K, 201K, 301K, 401K, and 501K steps, while **high-resolution sampling (57 checkpoints)** reveals the inherent noise and fluctuations in training. Despite significant training noise, the overall semantic development follows a stable, predictable pattern captured by the representative sampling, as shown in Figure 16. The complete training trajectory with all 57 checkpoints is presented in Figure 17.

Moreover, this double-phase dynamics occurs when testing attention sparsity, effective rank, and $\mathcal{L}$ values (Figures 18, 19). Meaning, all of which exhibit the same early rapid shift followed by a slower restructuring phase. This convergence across independent metrics indicates that the model is not merely improving categorical alignment, but reorganizing its internal representations toward increasingly efficient structure.

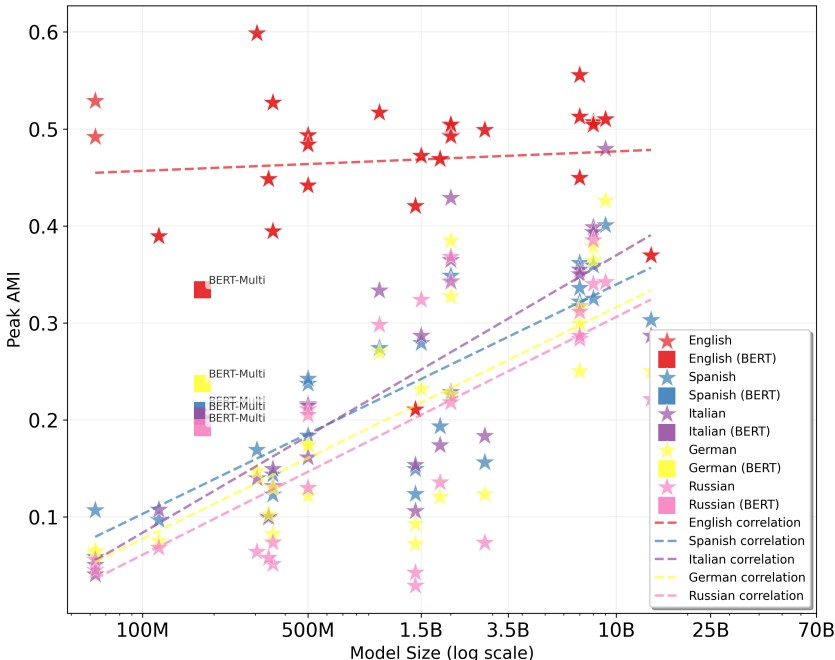

Figure 11: **Scale effect emerges for non-English languages.** We compute peak AMI as a function of model size across different languages: English (red), Spanish (blue), Italian (purple), German (yellow), and Russian (pink). While English shows no systematic scaling effect, the other languages exhibit a clear positive relationship between model size and AMI. We interpret this as a consequence of limited non-English training data: larger models are more likely to have been exposed to sufficient multilingual data, improving their conceptual alignment. Additional analyses across other RQs using the multilingual data support this hypothesis.

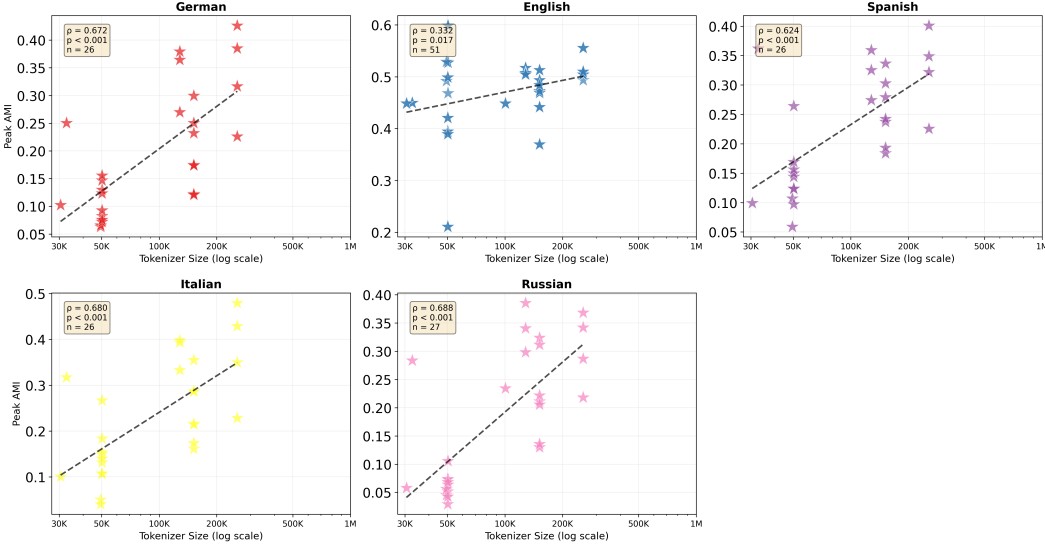

Figure 12: **Scale effect emerges for non-English languages.** We compute peak AMI as a function of model size across different languages: English (red), Spanish (blue), Italian (purple), German (yellow), and Russian (pink). While English shows no systematic scaling effect, the other languages exhibit a clear positive relationship between model size and AMI. We interpret this as a consequence of limited non-English training data: larger models are more likely to have been exposed to sufficient multilingual data, improving their conceptual alignment. Additional analyses across other RQs using the multilingual data support this hypothesis.

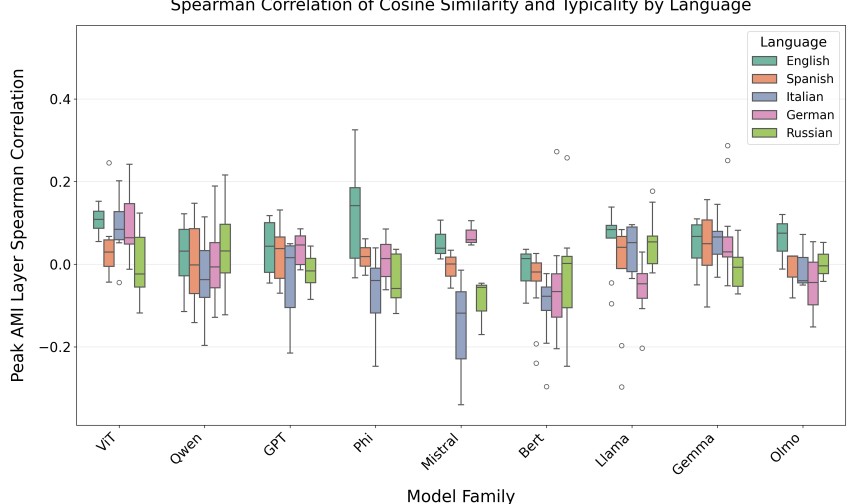

Figure 13: **Preservation of internal conceptual geometry across languages.** Models struggle to maintain the internal structure of human concepts in all tested languages, with similar patterns observed across English, Spanish, German, Italian, and Russian.

## I  DATASETS POLYSEMY

**Scope.** We quantify lexical ambiguity in our psycholinguistic stimuli by counting the distinct Word-Net synsets associated with each lemma. This *polysemy score* lets us estimate how many alternative senses a model must implicitly conflate when it produces a single embedding for a word.

**Why it matters.** Consider *bat*, which can denote either a flying mammal or a piece of sports equipment. The same vector must account for both senses. Aggregating semantically distant senses can blur the representation and thus confound model-human comparisons, especially in tasks that rely on fine-grained semantic similarity. Explicitly tracking polysemy allows us to verify that any performance effects we observe are not artefacts of lexical ambiguity.

**Results** Figure 20 shows the distribution of polysemy scores. The majority of items are unambiguous (1-2 senses), but a heavy-tailed minority (e.g. *Running* (52 senses), *Saw* (28), *Block* (28)) is highly polysemous. This suggests that our findings are due to real differences between the models rather than polysemy-related artifacts. An additional 141 lemmas that are not in WordNet were omitted.

## J  TOKENIZER ANALYSIS

**Rational.** The tokenizer of a model has a significant influence over the representations: segmentation rules (WordPiece vs. BPE), vocabulary size and special control tokens can inflate sequence length, skew frequency statistics, and shape error patterns. To ensure fair cross-model comparisons, we therefore (i) cluster checkpoints by the *tokenizer they use* and (ii) quantify how much those tokenizers overlap when applied to our datasets.

**Procedure.** Before computing overlap, we normalize the vocabulary, stripping tokenizer-specific characters; SentencePiece prefixes (‗), GPT-style BPE space prefixes (Ġ/ġ) and newline markers (Ċ), WordPiece continuations markers (##), and related block characters. After this cleanup, tokens differing only by such prefixes collapse to a shared canonical form (e.g. ‗house, Ġhouse, and ##house all become house). We then compute pair-wise Jaccard similarity on these cleaned vocabularies.

Table 1 summarizes the core statistics and information regarding the tokenizer types, while Figure 21 visualizes the resulting pairwise vocabulary overlap.

**Findings.** We find that most tokenizer families share substantial lexical overlap, often exceeding 60%, suggesting a de-facto common token inventory across recent open-source models. First-generation BERT WordPiece (bert-large-uncased and bert-base-uncased) are an outlier, sharing under 16% of tokens with any other group.

| Model Family | Mean Tokens / Item | Vocabulary Size | Tokenizer Type |
|---|---|---|---|
| BERT | 1.65 | 30K | WordPiece |
| DeBERTa & RoBERTa | 2.30 | 50K | WordPiece |
| GPT | 2.30 | 50K | BPE |
| Gemma | 1.65 | 256K | SentencePiece (subword) |
| Llama | 2.19 | 128K | BPE (SentencePiece/Tiktoken) |
| Mistral | 2.35 | 32K | BPE + control tokens |
| Phi | 2.30 | 32K | BPE (SentencePiece/Tiktoken) |
| Qwen | 2.19 | 151.6K | BPE |

Table 1: **Tokenizer statistics by tokenizer family.** Mean Tokens/Items refer to the average of the tokens per item in our datasets. The columns Vocabulary Size and Tokenizer Type are properties of the tokenizer.

**Model Clustering By Tokenizer Family**

- **Llama:** Llama-3.2-1B (representative), Llama-3.1-8B, Meta-Llama-3-8B, Llama-3.1-70B, Meta-Llama-3-70B
- **Gemma:** gemma-2-9b (representative), gemma-7b, gemma-2b, gemma-2-2b
- **Mistral:** Mistral-7B-v0.3 (unique)
- **Phi:** phi-2 (representative), phi-1, phi-1.5
- **RoBERTa:** roberta-large (unique)
- **DeBERTa:** deberta-large (unique)
- **GPT**: gpt2-medium (representative), gpt2
- **BERT:** bert-large-uncased (representative), bert-base-uncased
- **Qwen:** Qwen1.5-0.5B (representative), Qwen1.5-1.8B, Qwen1.5-14B, Qwen2-0.5B, Qwen2-7B, Qwen2.5-0.5B, Qwen2.5-1.5B, Qwen2.5-32B, Qwen1.5-32B, phi-4[3]

## K  ADDITIONAL CLUSTERING METRICS

To further validate our cluster alignment findings (Section 5.1), in addition to Adjusted Mutual Information (AMI) and the Normalized Mutual Information (NMI), we also computed the Adjusted Rand Index (ARI) for the k-means clusters derived from LLM embeddings against human-defined categories. ARI measures the similarity between two data clusterings, correcting for chance. Like AMI, a score of 1 indicates perfect agreement and 0 indicates chance agreement.

Across all tested LLMs, the ARI and NMI scores largely mirrored the trends observed with AMI, showing significantly above-chance alignment with human categories and similar relative model performances. Silhouette scores, while more variable, generally indicated reasonable cluster cohesion for both LLM-derived and human categories. Detailed tables of these scores are provided below.

These supplementary metrics reinforce the conclusion that LLMs capture broad human-like conceptual groupings.

## L  MINI-CONTROLLED EXPERIMENT (MATCHED TRAINING DATA)

To evaluate the extent to which dataset differences might account for the architectural patterns we report, we conducted matched-family analyses involving the only model families that can be aligned

---

[3]phi-4 has a different tokenizer than the rest of Phi family. The results of its tokenizer match the tokenizer of the Qwen family.

in both training data: GPT, Pythia, Cerebras, and T5. While these comparisons cannot rule out all confounds, they substantially reduce the influence of dataset variation. Across all matched settings, encoder models continue to outperform decoder models, yielding higher AMI and lower $\mathcal{L}$. This indicates that the architectural effects observed throughout the paper cannot be explained by differences in training data alone.

## M   DETAILED AMI SCORES PER MODEL AND DATASET

Table 2 provides a more granular view of the static AMI scores for each LLM across the three individual psychological datasets.

| Dataset | Model | NMI | AMI | ARI |
|---|---|---|---|---|
| (Rosch, 1973c) | bert-large-uncased | 0.19453 | 0.2011 | 0.11336 |
| (Rosch, 1975) | bert-large-uncased | 0.16547 | 0.27324 | 0.2216 |
| (McCloskey & Glucksberg, 1978) | bert-large-uncased | 0.12003 | 0.15934 | 0.06306 |
| (Rosch, 1973c) | FacebookAI/roberta-large | 0.1021 | 0.10666 | 0.03393 |
| (Rosch, 1975) | FacebookAI/roberta-large | 0.12138 | 0.23938 | 0.14165 |
| (McCloskey & Glucksberg, 1978) | FacebookAI/roberta-large | 0.06271 | 0.08873 | 0.03173 |
| (Rosch, 1973c) | google-t5/t5-large | 0.16583 | 0.16855 | 0.03676 |
| (Rosch, 1975) | google-t5/t5-large | -0.03799 | 0.04179 | 0.00758 |
| (McCloskey & Glucksberg, 1978) | google-t5/t5-large | 0.06146 | 0.08825 | 0.0082 |
| (Rosch, 1973c) | google/gemma-2-27b | 0.08523 | 0.09065 | 0.04158 |
| (Rosch, 1975) | google/gemma-2-27b | 0.04276 | 0.10062 | 0.06244 |
| (McCloskey & Glucksberg, 1978) | google/gemma-2-27b | 0.07814 | 0.10274 | 0.04364 |
| (Rosch, 1973c) | google/gemma-2-2b | 0.04029 | 0.04107 | 0.01212 |
| (Rosch, 1975) | google/gemma-2-2b | 0.04529 | 0.14844 | 0.07596 |
| (McCloskey & Glucksberg, 1978) | google/gemma-2-2b | 0.09953 | 0.13593 | 0.06326 |
| (Rosch, 1973c) | google/gemma-2-9b | 0.1222 | 0.12757 | 0.06053 |
| (Rosch, 1975) | google/gemma-2-9b | 0.07841 | 0.16126 | 0.09617 |
| (McCloskey & Glucksberg, 1978) | google/gemma-2-9b | 0.10879 | 0.13997 | 0.06439 |
| (Rosch, 1973c) | google/gemma-2b | 0.04336 | 0.04616 | 0.01593 |
| (Rosch, 1975) | google/gemma-2b | -0.00353 | 0.04483 | 0.01577 |
| (McCloskey & Glucksberg, 1978) | google/gemma-2b | 0.03472 | 0.05484 | 0.02142 |
| (Rosch, 1973c) | google/gemma-7b | 0.04459 | 0.04547 | 0.01052 |
| (Rosch, 1975) | google/gemma-7b | -0.03055 | 0.02644 | 0.01506 |
| (McCloskey & Glucksberg, 1978) | google/gemma-7b | 0.03338 | 0.05724 | 0.02176 |
| (Rosch, 1973c) | meta-llama/Llama-3.1-70B | 0.03008 | 0.03528 | 0.01936 |
| (Rosch, 1975) | meta-llama/Llama-3.1-70B | -0.07026 | 0.02636 | 0.00392 |
| (McCloskey & Glucksberg, 1978) | meta-llama/Llama-3.1-70B | -0.04773 | 0.00972 | 0.00236 |
| (Rosch, 1973c) | meta-llama/Llama-3.1-8B | 0.00473 | 0.00393 | 0.00023 |
| (Rosch, 1975) | meta-llama/Llama-3.1-8B | -0.03928 | 0.05489 | 0.01884 |
| (McCloskey & Glucksberg, 1978) | meta-llama/Llama-3.1-8B | -0.02671 | 0.02208 | 6.00E-05 |
| (Rosch, 1973c) | meta-llama/Llama-3.2-1B | 0.01936 | 0.01567 | 0.00246 |
| (Rosch, 1975) | meta-llama/Llama-3.2-1B | -0.01876 | 0.05663 | 0.00782 |
| (McCloskey & Glucksberg, 1978) | meta-llama/Llama-3.2-1B | 0.03625 | 0.06798 | 0.01352 |
| (Rosch, 1973c) | meta-llama/Llama-3.2-3B | 0.03757 | 0.03537 | 0.00876 |
| (Rosch, 1975) | meta-llama/Llama-3.2-3B | 0.01893 | 0.09619 | 0.03193 |
| (McCloskey & Glucksberg, 1978) | meta-llama/Llama-3.2-3B | 0.03914 | 0.07395 | 0.0202 |
| (Rosch, 1973c) | meta-llama/Meta-Llama-3-70B | 0.02289 | 0.03133 | 0.01514 |
| (Rosch, 1975) | meta-llama/Meta-Llama-3-70B | -0.06428 | 0.0185 | 0.00554 |
| (McCloskey & Glucksberg, 1978) | meta-llama/Meta-Llama-3-70B | -0.04595 | 0.01068 | 0.00272 |
| (Rosch, 1973c) | meta-llama/Meta-Llama-3-8B | 0.03512 | 0.02852 | 0.00225 |
| (Rosch, 1975) | meta-llama/Meta-Llama-3-8B | -0.06011 | 0.03694 | 0.00676 |
| (McCloskey & Glucksberg, 1978) | meta-llama/Meta-Llama-3-8B | -0.0355 | 0.0219 | 0.00676 |
| (Rosch, 1973c) | microsoft/deberta-large | 0.03748 | 0.03909 | 0.01467 |
| (Rosch, 1975) | microsoft/deberta-large | 0.16568 | 0.28993 | 0.20527 |
| (McCloskey & Glucksberg, 1978) | microsoft/deberta-large | 0.03217 | 0.06175 | 0.03019 |
| (Rosch, 1973c) | microsoft/phi-1_5 | 0.02102 | 0.01786 | 0.0075 |
| (Rosch, 1975) | microsoft/phi-1_5 | 0.03989 | 0.13887 | 0.04305 |
| (McCloskey & Glucksberg, 1978) | microsoft/phi-1_5 | 0.00895 | 0.05215 | 0.00639 |

| (Rosch, 1973c) | microsoft/phi-1 | 0.0249 | 0.01698 | 0.00133 |
|---|---|---|---|---|
| (Rosch, 1975) | microsoft/phi-1 | -0.03625 | 0.02811 | 0.00217 |
| (McCloskey & Glucksberg, 1978) | microsoft/phi-1 | -0.01148 | 0.03085 | 0.00371 |
| (Rosch, 1973c) | microsoft/phi-2 | 0.03703 | 0.02968 | 0.00404 |
| (Rosch, 1975) | microsoft/phi-2 | -0.03654 | 0.04227 | 0.03942 |
| (McCloskey & Glucksberg, 1978) | microsoft/phi-2 | -0.00254 | 0.02531 | 0.00533 |
| (Rosch, 1973c) | microsoft/phi-4 | 0.03075 | 0.03043 | 0.01076 |
| (Rosch, 1975) | microsoft/phi-4 | -0.06737 | 0.00092 | -0.01361 |
| (McCloskey & Glucksberg, 1978) | microsoft/phi-4 | -0.01789 | 0.02705 | 0.00066 |
| (Rosch, 1973c) | mistralai/Mistral-7B-v0.3 | 0.0425 | 0.03507 | 0.00357 |
| (Rosch, 1975) | mistralai/Mistral-7B-v0.3 | -0.05018 | 0.01217 | 0.0177 |
| (McCloskey & Glucksberg, 1978) | mistralai/Mistral-7B-v0.3 | -0.01264 | 0.03902 | 0.00931 |
| (Rosch, 1973c) | Qwen/Qwen1.5-0.5B | 0.00148 | -0.00225 | 0.00399 |
| (Rosch, 1975) | Qwen/Qwen1.5-0.5B | -0.01538 | 0.04833 | 0.0095 |
| (McCloskey & Glucksberg, 1978) | Qwen/Qwen1.5-0.5B | 0.02559 | 0.06023 | 0.00771 |
| (Rosch, 1973c) | Qwen/Qwen1.5-1.8B | 0.03397 | 0.03232 | 0.01034 |
| (Rosch, 1975) | Qwen/Qwen1.5-1.8B | -0.01129 | 0.05803 | 0.00683 |
| (McCloskey & Glucksberg, 1978) | Qwen/Qwen1.5-1.8B | -0.00541 | 0.03614 | 0.00538 |
| (Rosch, 1973c) | Qwen/Qwen1.5-14B | 0.0372 | 0.02738 | 0.0028 |
| (Rosch, 1975) | Qwen/Qwen1.5-14B | -0.02604 | 0.05153 | 0.01211 |
| (McCloskey & Glucksberg, 1978) | Qwen/Qwen1.5-14B | 0.00124 | 0.04136 | 0.00338 |
| (Rosch, 1973c) | Qwen/Qwen1.5-32B | 0.02638 | 0.02436 | 0.00409 |
| (Rosch, 1975) | Qwen/Qwen1.5-32B | -0.03413 | 0.02526 | -0.00665 |
| (McCloskey & Glucksberg, 1978) | Qwen/Qwen1.5-32B | -0.01991 | 0.02124 | -0.00059 |
| (Rosch, 1973c) | Qwen/Qwen1.5-4B | 0.03803 | 0.04058 | 0.01742 |
| (Rosch, 1975) | Qwen/Qwen1.5-4B | -0.03309 | 0.03988 | 0.01678 |
| (McCloskey & Glucksberg, 1978) | Qwen/Qwen1.5-4B | -0.03997 | 0.00548 | -0.00028 |
| (Rosch, 1973c) | Qwen/Qwen1.5-72B | 0.03697 | 0.02892 | 0.00144 |
| (Rosch, 1975) | Qwen/Qwen1.5-72B | -0.06184 | 0.02213 | 0.0017 |
| (McCloskey & Glucksberg, 1978) | Qwen/Qwen1.5-72B | -0.02022 | 0.02918 | 0.00297 |
| (Rosch, 1973c) | Qwen/Qwen2-0.5B | 0.02266 | 0.01923 | 0.00662 |
| (Rosch, 1975) | Qwen/Qwen2-0.5B | 0.0515 | 0.14571 | 0.04999 |
| (McCloskey & Glucksberg, 1978) | Qwen/Qwen2-0.5B | 0.01508 | 0.04357 | 0.00643 |
| (Rosch, 1973c) | Qwen/Qwen2-1.5B | 0.02956 | 0.02779 | 0.00544 |
| (Rosch, 1975) | Qwen/Qwen2-1.5B | -0.03595 | 0.03443 | -0.01099 |
| (McCloskey & Glucksberg, 1978) | Qwen/Qwen2-1.5B | 0.01768 | 0.05407 | 0.01604 |
| (Rosch, 1973c) | Qwen/Qwen2-7B | 0.06424 | 0.06439 | 0.02067 |
| (Rosch, 1975) | Qwen/Qwen2-7B | 0.0333 | 0.09155 | 0.02832 |
| (McCloskey & Glucksberg, 1978) | Qwen/Qwen2-7B | 0.05329 | 0.07599 | 0.01977 |
| (Rosch, 1973c) | Qwen/Qwen2.5-0.5B | 0.03165 | 0.03291 | 0.01029 |
| (Rosch, 1975) | Qwen/Qwen2.5-0.5B | -0.06534 | -0.0196 | -0.01165 |
| (McCloskey & Glucksberg, 1978) | Qwen/Qwen2.5-0.5B | 0.0062 | 0.04191 | 0.0054 |
| (Rosch, 1973c) | Qwen/Qwen2.5-1.5B | 0.04838 | 0.0489 | 0.0129 |
| (Rosch, 1975) | Qwen/Qwen2.5-1.5B | 0.03785 | 0.113 | 0.02761 |
| (McCloskey & Glucksberg, 1978) | Qwen/Qwen2.5-1.5B | 0.06166 | 0.08675 | 0.03162 |
| (Rosch, 1973c) | Qwen/Qwen2.5-3B | 0.03882 | 0.0348 | 0.00465 |
| (Rosch, 1975) | Qwen/Qwen2.5-3B | 0.03977 | 0.10821 | 0.04302 |
| (McCloskey & Glucksberg, 1978) | Qwen/Qwen2.5-3B | 0.03416 | 0.07307 | 0.02959 |
| (Rosch, 1973c) | Qwen/Qwen2.5-7B | 0.0529 | 0.05051 | 0.01605 |
| (Rosch, 1975) | Qwen/Qwen2.5-7B | -0.00905 | 0.03227 | 0.01044 |
| (McCloskey & Glucksberg, 1978) | Qwen/Qwen2.5-7B | 0.00222 | 0.02759 | 0.00551 |

Table 2: Mutual information measures (normalized mutual information, adjusted mutual information, adjusted rand index) per model per dataset. Aggregated results are shown in the main paper and the Figures in the Appendix.

# N    CORRELATION BETWEEN HUMAN TYPICALITY JUDGMENTS AND LLM INTERNAL CLUSTER GEOMETRY

The following tables present the Spearman correlation coefficients ($\rho$) between human typicality judgments and LLM internal representations across different analysis approaches:

**Table 3**: Static analysis correlations using embeddings from the E matrix. This approach captures the baseline semantic relationships between items and categories without contextual processing.

**Table 4**: Peak AMI layer analysis correlations using contextual embeddings from the layer that maximized AMI scores (as identified in RQ1). This approach leverages the optimal layer for semantic clustering to assess fine-grained semantic fidelity.

Both tables present correlations across three cognitive science datasets: Rosch (1973), Rosch (1975), and McCloskey (1978), with asterisks (*) indicating statistically significant correlations ($p < 0.05$). The modest correlation values across most models suggest limited alignment between LLM internal representations and human-perceived semantic nuances.

| Model | Dataset Correlation (Spearman $\rho$) - Static Layer | | |
| --- | --- | --- | --- |
| | **Rosch (1973)** | **Rosch (1975)** | **McCloskey (1978)** |
| **Deberta large (304M)** | 0.144 | 0.107* | 0.075 |
| **Bert large (340M)** | 0.378* | 0.275* | 0.250* |
| **Roberta large (355M)** | 0.005 | 0.038 | -0.029 |
| **Gemma (2B)** | 0.069 | 0.078 | 0.007 |
| **Gemma 2 (2B)** | 0.236 | 0.119* | 0.147* |
| **Gemma (7B)** | 0.131 | 0.100* | 0.007 |
| **Gemma 2 (9B)** | 0.280 | 0.135* | 0.199* |
| **Gemma 2 (27B)** | 0.112 | 0.122* | 0.161* |
| **Qwen1.5 (0.5B)** | 0.175 | 0.076 | 0.096* |
| **Qwen2 (0.5B)** | 0.238 | 0.041 | 0.040 |
| **Qwen2.5 (0.5B)** | 0.212 | 0.027 | 0.037 |
| **Qwen2.5 (1.5B)** | 0.141 | 0.086* | 0.078 |
| **Qwen1.5 (1.8B)** | 0.172 | 0.134* | 0.154* |
| **Qwen2 (7B)** | 0.036 | 0.087* | 0.040 |
| **Qwen1.5 (14B)** | 0.154 | 0.086* | 0.108* |
| **Qwen1.5 (32B)** | -0.032 | 0.100* | 0.081 |
| **Qwen2.5 (32B)** | 0.035 | 0.105* | 0.084 |
| **Mistral v0.3 (7B)** | 0.076 | 0.152* | -0.009 |
| **Llama 3.2 (1B)** | 0.301* | 0.056 | 0.039 |
| **Llama 3 (8B)** | -0.002 | 0.099* | 0.080 |
| **Llama 3.1 (8B)** | 0.004 | 0.108* | 0.081 |
| **Llama 3 (70B)** | 0.148 | 0.161* | 0.155* |
| **Llama 3.1 (70B)** | 0.122 | 0.161* | 0.155* |
| **Phi 1 (1.42B)** | 0.071 | 0.052 | 0.054 |
| **Phi 1.5 (1.42B)** | -0.088 | 0.079 | 0.018 |
| **Phi 2 (2.78B)** | -0.056 | 0.044 | 0.024 |
| **Phi 4 (14.7B)** | 0.079 | 0.086* | 0.097* |
| **T5 Large (770M)** | 0.235 | 0.259* | 0.178* |
| **GPT-2 Medium (355M)** | -0.032 | 0.063 | -0.017 |
| **ViT-B/32 Text (63.1M)** | 0.527* | 0.315* | 0.286* |
| **ViT-B/16 Text (63.1M)** | 0.528* | 0.289* | 0.278* |
| **Word2Vec (300D)** | 0.442* | 0.349* | 0.437* |
| **Glove (300D)** | 0.315* | 0.333* | 0.350* |

Table 3: **Correlation between Human Typicality Judgments and LLM Internal Cluster Geometry.** Spearman static-layer rank correlations between human-rated psychological typicality/distance (higher human scores = less typical/more distant) and item-to-centroid cosine similarity (higher similarity = more central to LLM cluster). *$p < 0.05$.

| Model | Dataset Correlation (Spearman $\rho$) - Peak AMI Layer | | |
|---|---|---|---|
| | Rosch (1973) | Rosch (1975) | McCloskey (1978) |
| Deberta large (304M) | 0.277 | 0.107* | 0.126* |
| Bert large (340M) | -0.120 | 0.148* | 0.026 |
| Roberta large (355M) | -0.011 | 0.038 | -0.022 |
| Gemma (2B) | 0.127 | 0.092* | 0.039 |
| Gemma 2 (2B) | 0.034 | 0.139* | 0.103* |
| Gemma (7B) | 0.004 | -0.050 | 0.088 |
| Gemma 2 (9B) | 0.047 | 0.110* | 0.098* |
| Gemma 2 (27B) | -0.135 | 0.090* | -0.101* |
| Qwen1.5 (0.5B) | -0.025 | -0.064 | 0.122* |
| Qwen2 (0.5B) | 0.090 | 0.064 | -0.077 |
| Qwen2.5 (0.5B) | -0.012 | 0.121* | 0.072 |
| Qwen2.5 (1.5B) | -0.114 | 0.084* | -0.028 |
| Qwen1.5 (1.8B) | 0.092 | -0.044 | -0.004 |
| Qwen2 (7B) | 0.004 | 0.049 | -0.087 |
| Qwen1.5 (14B) | 0.032 | 0.091* | 0.043 |
| Qwen1.5 (32B) | 0.026 | 0.082 | 0.111* |
| Qwen2.5 (32B) | 0.045 | 0.079 | -0.060 |
| Mistral v0.3 (7B) | 0.013 | 0.039 | 0.107* |
| Llama 3.2 (1B) | 0.063 | 0.094* | 0.070 |
| Llama 3 (8B) | -0.045 | 0.138* | 0.084 |
| Llama 3.1 (8B) | -0.096 | 0.130* | 0.087 |
| Llama 3 (70B) | -0.108 | 0.023 | 0.050 |
| Phi 1 (1.42B) | 0.216 | 0.185* | -0.029 |
| Phi 1.5 (1.42B) | 0.162 | 0.098* | 0.015 |
| Phi 2 (2.78B) | 0.325* | 0.142* | -0.033 |
| Phi 4 (14.7B) | 0.079 | 0.129* | 0.007 |
| T5 Large (770M) | 0.219 | 0.226* | 0.282* |
| GPT-2 Small (117M) | -0.046 | 0.118* | 0.010 |
| GPT-2 Medium (355M) | 0.077 | 0.109* | -0.029 |
| ViT-B/32 Text (63.1M) | 0.128 | 0.055 | 0.152* |
| ViT-B/16 Text (63.1M) | 0.089 | 0.086* | 0.127* |
| Word2Vec (300D) | 0.442* | 0.349* | 0.437* |
| Glove (300D) | 0.315* | 0.333* | 0.350* |

Table 4: **Correlation between Human Typicality Judgments and LLM Internal Cluster Geometry.** Spearman peak-AMI layer rank correlations between human-rated psychological typicality/distance (higher human scores = less typical/more distant) and item-to-centroid cosine similarity (higher similarity = more central to LLM cluster). $^*p < 0.05$.

## O TYPICALITY AND COSINE SIMILARITY [RQ2]

Figure 25 shows representative scatter plots illustrating the relationship between human typicality scores (or psychological distances) and the LLM-derived item-centroid cosine similarities for selected categories and models. These plots visually demonstrate the often modest correlations discussed in Section 5.2.

Figure 26 shows the aggregated Spearman correlation across model families and datasets. These correlations are very weak and mostly non-significant.

## P THEORETICAL EXTREME CASE EXPLORATION FOR $\mathcal{L}$

In the case where $|C| = |X|$ (each data point is a cluster of size 1, so $|C_c| = 1 \; \forall c \in C$), then $H(X|C) = \frac{1}{|X|} \sum_{c \in C} 1 \cdot \log_2 1 = 0$. The distortion term $\sigma_c^2 = 0$ for each cluster as the item is its own centroid. Thus, $\mathcal{L} = I(X;C) + \beta \cdot 0 = H(X) - H(X|C) = H(X) = \log_2 |X|$. This represents the cost of encoding each item perfectly without any compression via clustering, and zero distortion.

In the case where $|C| = 1$ (one cluster $C_X$ contains all $|X|$ data points, so $|C_{C_X}| = |X|$), then $H(X|C) = \frac{1}{|X|}|X|\log_2|X| = \log_2|X|$. Thus, $I(X;C) = H(X) - H(X|C) = \log_2|X| - \log_2|X| = 0$. This represents maximum compression (all items are treated as one). The distortion term becomes $\beta \cdot \frac{1}{|X|}|X| \cdot \sigma_X^2 = \beta \cdot \sigma_X^2$, where $\sigma_X^2$ is the variance of all items $X$ with respect to the global centroid of $X$. So, $\mathcal{L} = 0 + \beta \cdot \sigma_X^2 = \beta \cdot \sigma_X^2$. This represents the scenario of maximum compression where the cost is purely the distortion incurred by representing all items by a single prototype.

## Q COMPRESSION FIGURES

Figure 28 depicts the IB-RDT objective ($\mathcal{L}$) vs. $K$. Lower $\mathcal{L}$ indicates a more optimal balance between compression ($I(X;C)$) and semantic fidelity (distortion). Human categories (fixed $K$) show higher $\mathcal{L}$ values.

## R COMPLEXITY-DISTORTION RATIO (ON THE IMPORTANCE OF $\beta$)

Figure 29 provides an additional sensitivity analysis in which we examine the ratio between the distortion and complexity components of $\mathcal{L}$ as $\beta$ varies. Across all three datasets, encoder models maintain flat profiles, indicating stable conceptual structure under compression, whereas decoder models exhibit stronger shifts, reflecting greater reallocation of representational capacity.

## S $\mathcal{L}$ OBJECTIVE VS. DOWNSTREAM TASK PERFORMANCE

Analysis of 13 instruction-tuned models across 5 families (Qwen, Llama, Gemma, Phi, Mistral; see Table 5 for results) indicates no statistical significance ($r = -0.202$, $p = 0.508$). This finding suggests that while the $\mathcal{L}$ objective successfully identifies models that compress semantic categories more effectively, this compression ability does not directly translate to improved performance on standard NLP benchmarks. The lack of correlation implies that concept compression and benchmark accuracy represent distinct aspects of model capability, with the former capturing semantic organization efficiency and the latter measuring general knowledge and reasoning abilities. We specifically chose instruction-tuned models to ensure fair comparison on MMLU, as base models would likely perform poorly on this instruction-following benchmark. While our analysis covers a diverse range of model families and sizes, this represents a subset of available models due to the limited availability of instruction-tuned variants.

| Model | Size | $\mathcal{L}$ Score | MMLU Score |
|---|---|---|---|
| Qwen2-0.5B-Instruct | 494M | 1.930 | 0.433 |
| Qwen2.5-0.5B-Instruct | 494M | 1.982 | 0.469 |
| Llama-3.2-1B-Instruct | 1.2B | 1.876 | 0.454 |
| Qwen2.5-1.5B-Instruct | 1.5B | 2.071 | 0.597 |
| Gemma-2B-IT | 2.5B | 2.382 | 0.366 |
| Gemma-2-2B-IT | 2.6B | 2.263 | 0.565 |
| Phi-4-mini-instruct | 3.8B | 1.905 | 0.678 |
| Mistral-7B-Instruct-v0.3 | 7.2B | 1.714 | 0.603 |
| Qwen2-7B-Instruct | 7.6B | 2.319 | 0.700 |
| Meta-Llama-3-8B-Instruct | 8.0B | 1.348 | 0.647 |
| Llama-3.1-8B-Instruct | 8.0B | 1.320 | 0.679 |
| Gemma-7B-IT | 8.5B | 2.372 | 0.512 |
| Gemma-2-9B-IT | 9.2B | 2.467 | 0.723 |

Table 5: **No correlation between $\mathcal{L}$ objective scores and MMLU scores across different sizes and families.** Table displays $\mathcal{L}$ objective values vs. MMLU scores for 13 instruction-tuned models. Correlation measured ($r = -0.202$, $p = 0.508$).

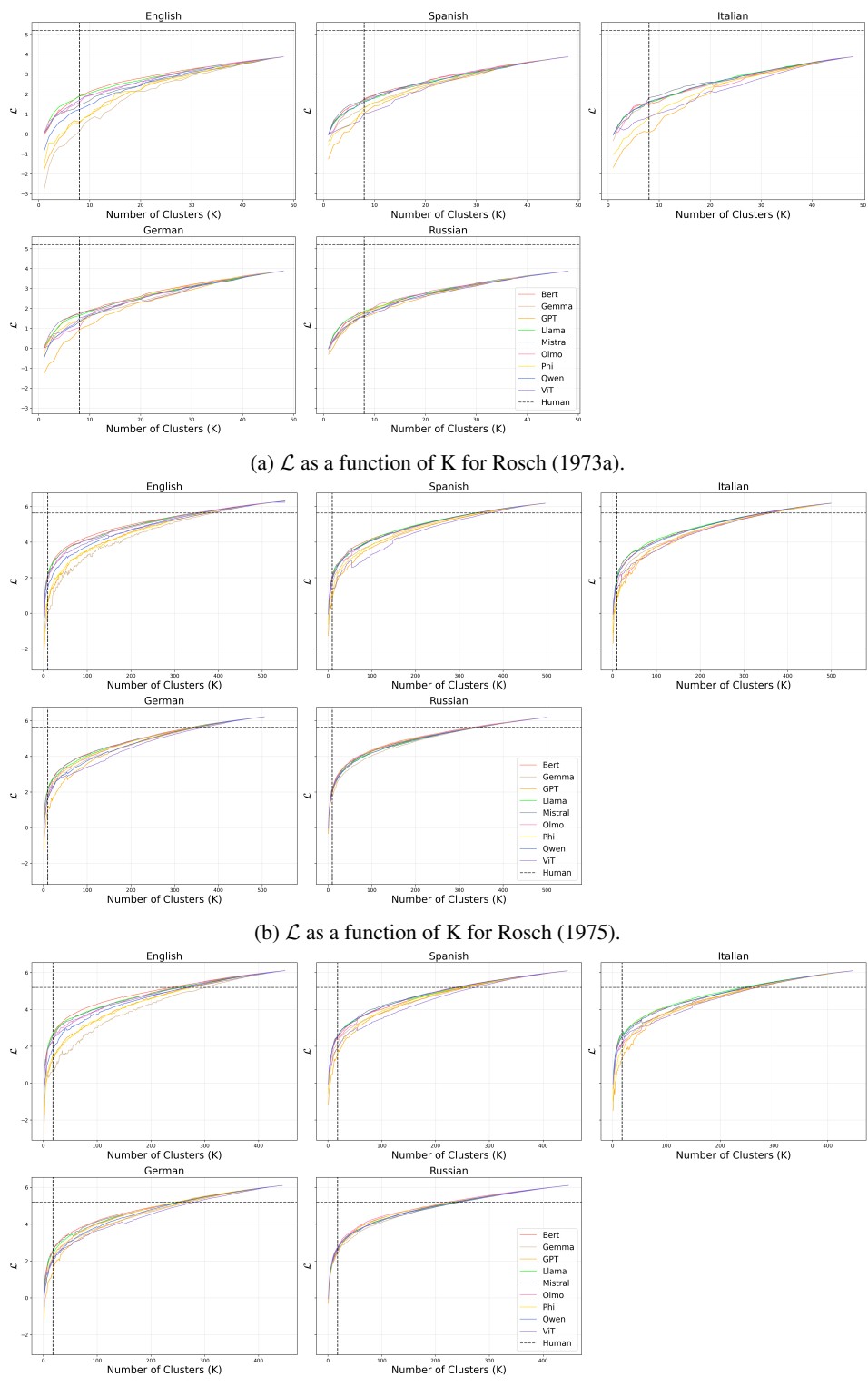

(a) $\mathcal{L}$ as a function of K for Rosch (1973a).

(b) $\mathcal{L}$ as a function of K for Rosch (1975).

(c) $\mathcal{L}$ as a function of K for McCloskey & Glucksberg (1978).

Figure 14: **Compression of non-English representations.** All non-English languages (Spanish, German, Italian, Russian) exhibit higher compression than English, with smaller models showing the greatest compression. This supports the interpretation that limited non-English training data leads to less flexible and less interpretable representations.

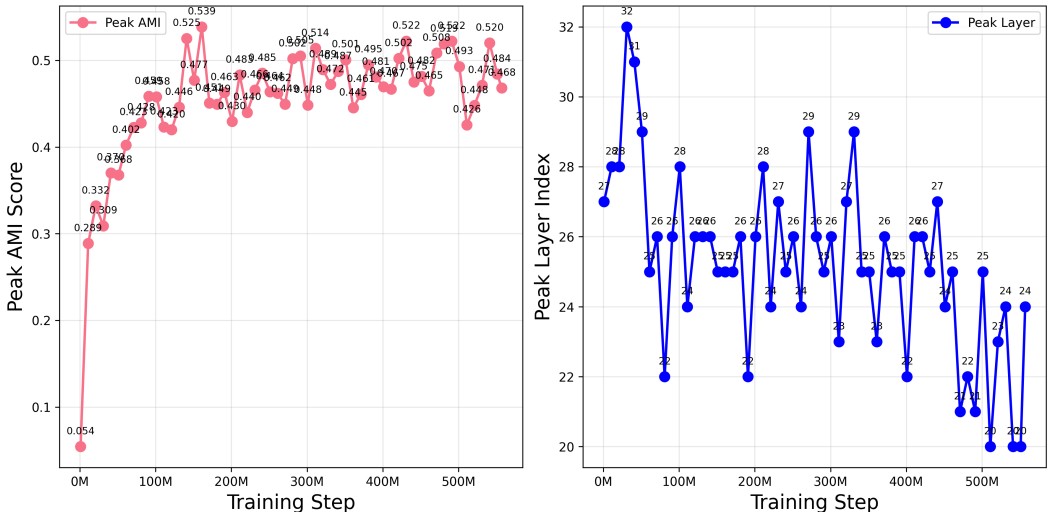

Figure 15: **OLMo-7B develops conceptual structure through two-phase dynamics.** *Left*: AMI with human categories rises rapidly in early training then refines gradually. *Right*: Peak semantic processing migrates from deep (layer 28) toward mid-network layers during training, revealing architectural reorganization for efficiency. Representative checkpoints shown; full 57-checkpoint analysis in Appendix B.5.

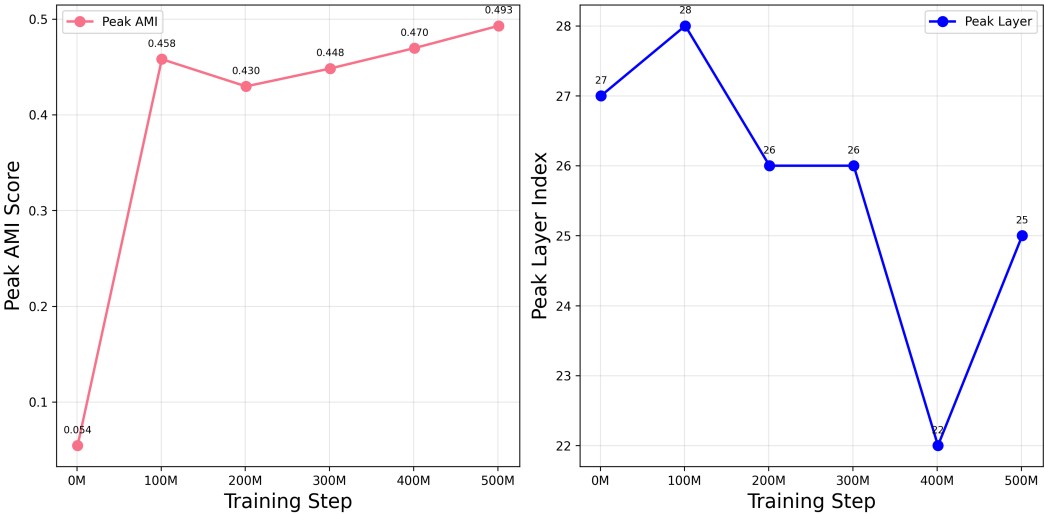

Figure 16: **Left: OLMo-7B representations steadily strengthen during training**: Concept representations develop rapidly at early steps, then refine more gradually over time. **Right: Semantic processing shifts from deep to mid-network layers**: The model undergoes a two-phase dynamic - initially moving semantic processing upward during rapid learning, then reorganizing to optimize efficiency while preserving performance. To improve readability, we present six representative checkpoints that capture the trend.

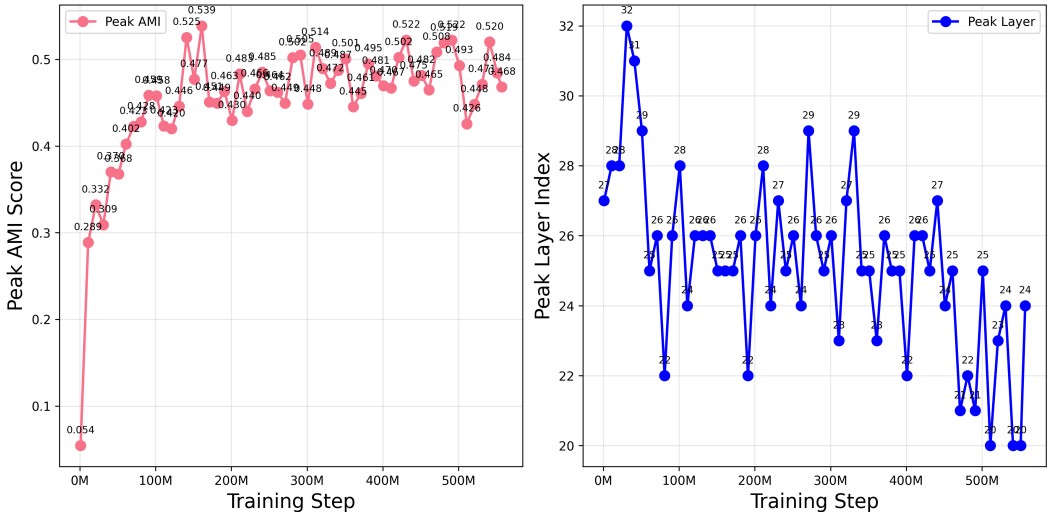

Figure 17: **Complete OLMo-7B training trajectory across 57 checkpoints**: This high-resolution view reveals the inherent noise and fluctuations in training, with individual checkpoint measurements varying throughout the process. Despite this variability, the overall trend aligns with the stable pattern shown in Figure 16, demonstrating that representative sampling effectively captures the underlying semantic development trajectory while filtering out training noise.

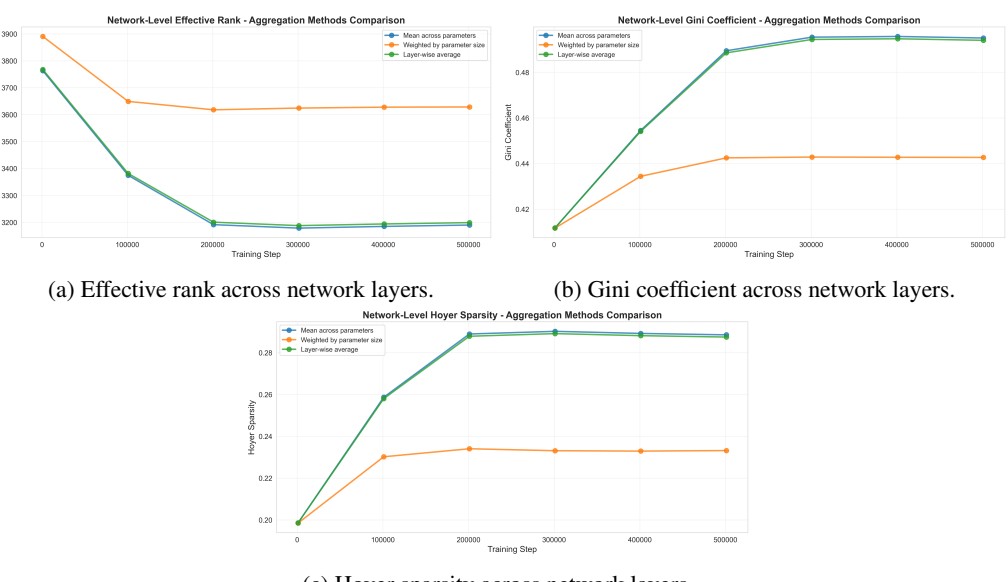

(a) Effective rank across network layers.     (b) Gini coefficient across network layers.

(c) Hoyer sparsity across network layers.

Figure 18: **Network-level measures of representational structure across training.** Each panel shows a different metric: (a) effective rank, (b) Gini coefficient, and (c) Hoyer sparsity computed across network layers. All three measures reveal the same two-phase developmental pattern observed in Figure 15: an early rapid change followed by slower restructuring, indicating coordinated reorganization of internal representations.

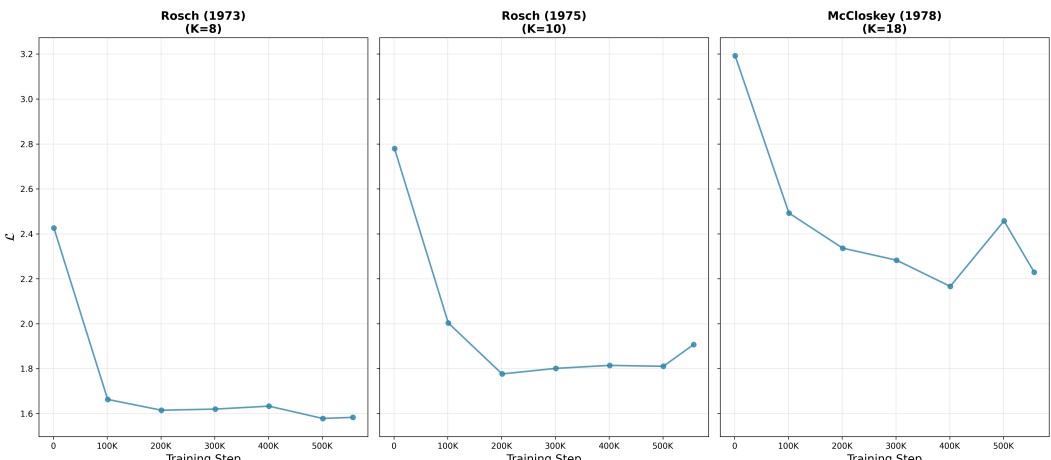

Figure 19: $\mathcal{L}$ **across training.** Across all three human benchmarks, Olmo training checkpoints reveal the same two-phase developmental pattern observed in Figure 15: an early rapid change followed by slower restructuring, indicating coordinated reorganization of internal representations.

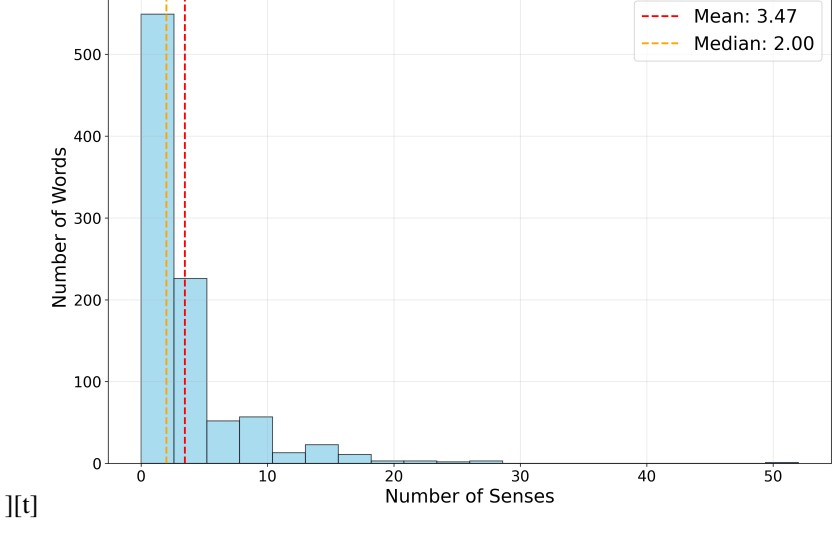

][t]

Figure 20: **Polysemy is not likely to influence our results as most of the words in our dataset are concrete nouns, which tend to be less polysemous.** Histogram of WordNet sense counts for the 943 lemmas in our benchmark. The dashed lines indicate the median and mean (2 senses, 3.47 senses, respectively).

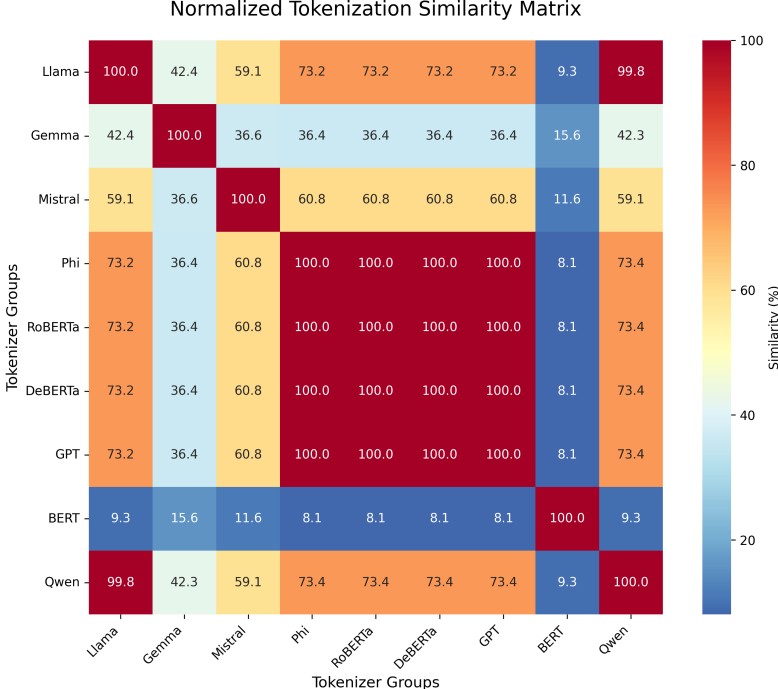

Figure 21: **Substantial lexical overlap suggests that tokenization differences alone cannot explain the observed performance variations in our experiments.** Vocabulary overlap between different tokenizers; most tokenizer families share substantial lexical overlap.

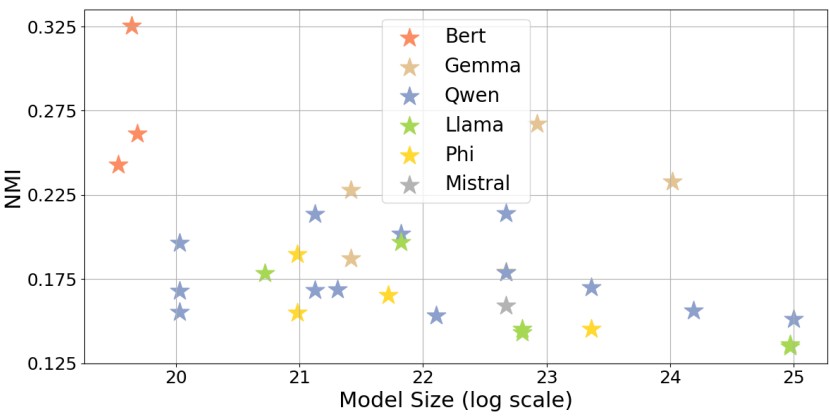

Figure 22: **LLM-derived Clusters Show Above-Chance Alignment with Human Conceptual Categories.** Normalized Mutual Information (NMI) between human-defined categories and clusters from static LLM embeddings. Results are averaged over three psychological datasets. All models perform significantly better than random clustering. BERT's performance is notably strong.

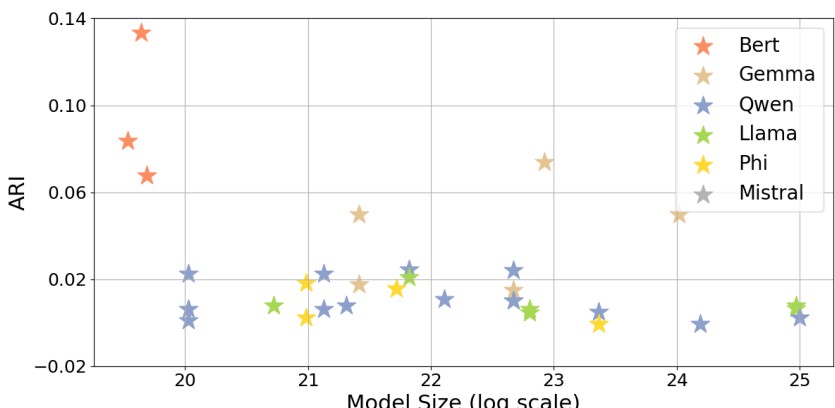

Figure 23: **LLM-derived Clusters Show Above-Chance Alignment with Human Conceptual Categories.** Adjusted Rand Index (ARI) between human-defined categories and clusters from LLM static embeddings. Results are averaged over three psychological datasets. All models perform significantly better than random clustering. BERT's performance is notably strong.

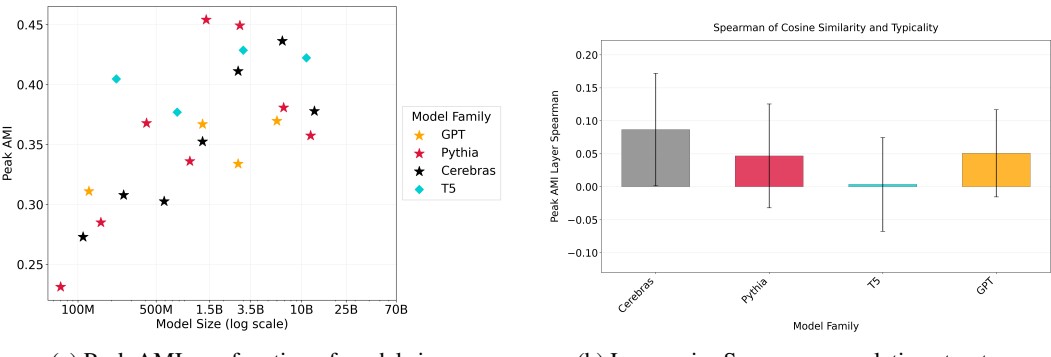

(a) Peak AMI as a function of model size.          (b) Layer-wise Spearman correlation structure.

Figure 24: **Evidence that encoder-decoder differences are not driven by dataset artifacts.** We compare the only model families that can be matched in training data GPT, Pythia, Cerebras, and T5. While these comparisons cannot eliminate all possible confounds, they show that the architectural patterns we report are robust: encoder models consistently achieve higher AMI and lower $L$, suggesting that the observed differences cannot be explained by dataset variation alone.

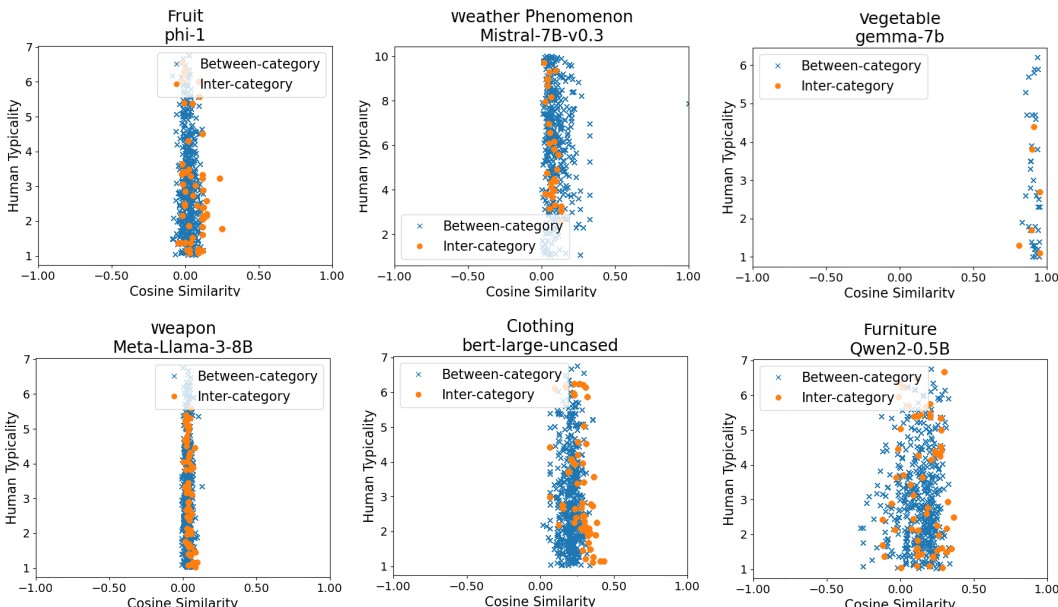

Figure 25: **Weak-to-No Correlation Between LLM Embedding Distance and Human Typicality Judgments.** Scatter plot examples of the cosine similarity versus the human typicality of items belonging to the category compared to items from other categories.

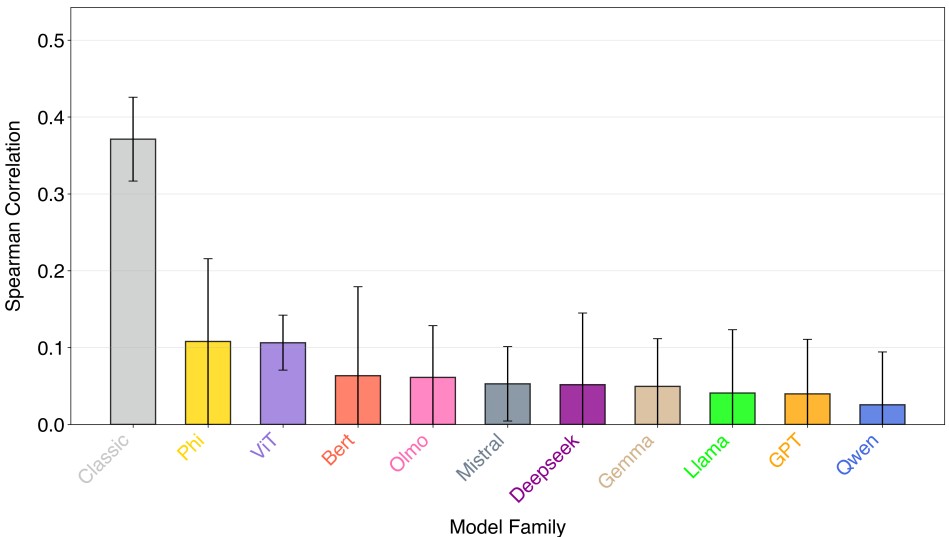

Figure 26: **Weak and Mostly Non-Significant Spearman Correlation Values Between Human Typicality Judgments and LLM Cosine Similarity Indicating Different Structure Representing Concepts.** Mean Static Layer Spearman correlation values across the models belonging to the same family and across the three datasets.

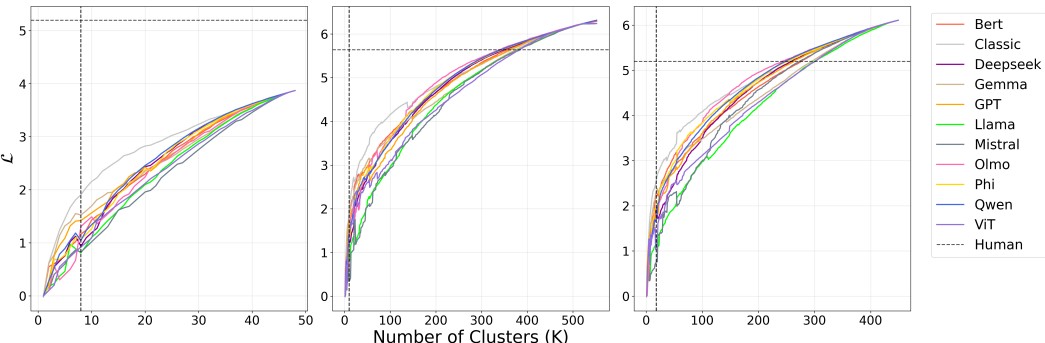

Figure 27: **Static Embeddings Achieve a more "Optimal" Compression-Meaning Trade-off by the $\mathcal{L}$ Measure.** IB-RDT objective ($\mathcal{L}$) vs. $K$ across all datasets. Lower $\mathcal{L}$ indicates a more optimal balance between compression ($I(X;C)$) and semantic fidelity (distortion). Static embeddings consistently achieve lower $\mathcal{L}$ values than both human categories and contextual embeddings. The plots correspond to the three datasets in the following order: Rosch (1973a), Rosch (1975), McCloskey & Glucksberg (1978).

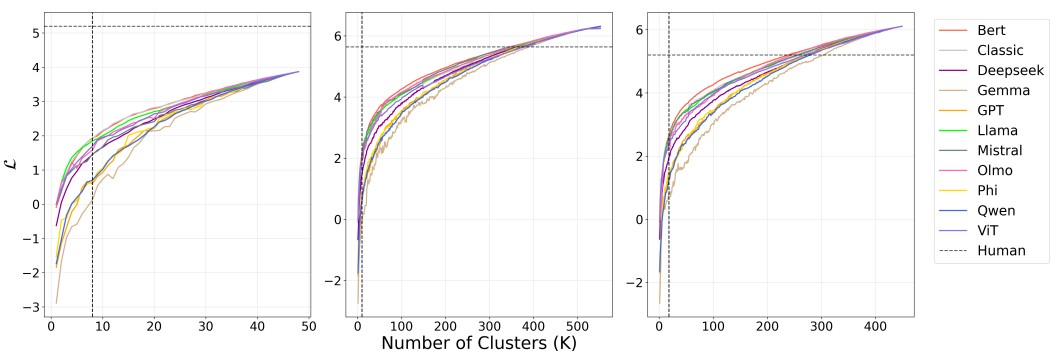

Figure 28: **Contextual Embeddings Achieve Better-than-Human Compression-Meaning Trade-off by the $\mathcal{L}$ Measure.** IB-RDT objective ($\mathcal{L}$) vs. $K$ across all datasets. Lower $\mathcal{L}$ indicates a more optimal balance between compression ($I(X;C)$) and semantic fidelity (distortion). Contextual embeddings outperform human categories but achieve higher $\mathcal{L}$ values than static embeddings. The plots correspond to the three datasets in the following order: Rosch (1973a), Rosch (1975), McCloskey & Glucksberg (1978).

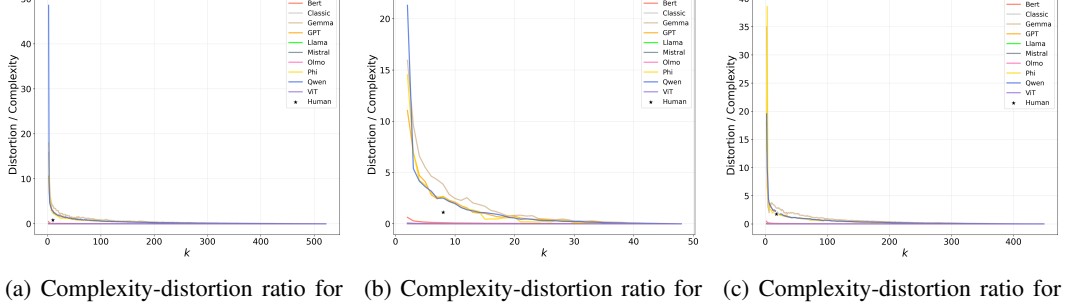

(a) Complexity-distortion ratio for Rosch (1973a).

(b) Complexity-distortion ratio for Rosch (1975).

(c) Complexity-distortion ratio for McCloskey & Glucksberg (1978).

Figure 29: **Complexity-Distortion Ratios Show That Encoder-Models are Less Sensitive to Variations in $\beta$.** For each dataset, we compute the ratio between distortion (loss of human-aligned conceptual information) and complexity (representation size) across values of the rate-distortion tradeoff parameter $\beta$. Encoder models yield consistently flatter profiles, indicating that their token embeddings preserve conceptual structure even under increasing compression. Decoder models exhibit more pronounced shifts, suggesting that they redistribute representational capacity more aggressively as $\beta$ increases, leading to higher sensitivity in this tradeoff.

