# OpenReview forum: "From Tokens to Thoughts: How LLMs and Humans Trade Compression for Meaning"
_ICLR.cc/2026/Conference — ICLR 2026 Poster_

### Official Review · Reviewer_X8mp · 2025-10-29

**Soundness:** 3
**Presentation:** 3
**Contribution:** 3
**Rating:** 6
**Confidence:** 3

**Summary:**

The paper studies how LLMs and humans trade-off compression and meaning in conceptual representations. Building on Rate–Distortion Theory and the Information Bottleneck (IB), the authors propose an objective function to combines information-theoretic compression with geometric coherence. Using digitized, classic human categorization benchmarks and embeddings from many models (encoders and decoders), they report three core findings: (i) LLMs broadly align with human category boundaries, (ii) LLMs capture weak typicality gradients relative to humans, and (iii) LLM-derived clusters achieve lower L (greater compression efficiency) than human categories, suggesting humans preserve more semantic richness at the expense of statistical efficiency. The paper releases the digitized benchmarks and positions the framework as a diagnostic for monitoring compression–meaning balance.

**Strengths:**

1. Originality. A clear, unified information-theoretic lens that ties together clustering complexity (compression) and semantic coherence (meaning), then uses it to compare humans and LLMs at scale. The encoder-decoder contrast is especially thought-provoking.
2. Quality. Careful benchmark curation (classic cognitive datasets), comprehensive model coverage, and multi-angle evaluation (category alignment, typicality, and L-curves); The training-dynamics analysis with 57 OLMo-7B checkpoints is helpful.
3. Clarity. The paper’s structure is easy to follow; figures that separate boundary alignment vs. typicality vs. L-frontiers help the reader parse what “efficiency” means in practice.
4. Significance. The finding that humans are less efficient but more semantically rich challenges a common “optimality” narrative and invites new objectives/architectures for more human-aligned understanding.

**Weaknesses:**

1. Many correlations are modest. Please provide bootstrap confidence intervals, multiple-comparison controls, and seed variability for clustering and correlation estimates.
2. The conclusion that humans are statistically suboptimal depends on the chosen geometry and L weighting. Most results fix $\beta=1$; the paper should report sensitivity curves over $\beta$ and discuss how human/LLM rankings vary under plausible trade-off weights. Also clarify how human distortion is computed from typicality/membership data to avoid mismatched metrics.
3. Encoder-decoder gap interpretation. The architecture result is intriguing, but confounded by training objective differences (MLM vs. autoregressive), tokenizers, and pretraining corpora. A controlled study with matched data/tokenizers and frozen-head probes would strengthen the claim.

**Questions:**

1. How exactly is the Distortion term computed for human categories, from typicality distances, membership uncertainty, or an inferred geometry? Please add a short methodological box that maps human ratings to the variance term and discuss limitations.
2. If typicality is computed against category names, can you replicate with prototype centroids (learned per category) or descriptive definitions, and do encoder-decoder gaps shrink?
3. With a common tokenizer/corpus and matched parameter counts, do masked-LM encoders still outperform autoregressive decoders on AMI and typicality?
4. During the mid-layer migration phase, what happens to the complexity-distortion frontier layer-wise? Does the “efficiency” shift track changes in heads/FFNs (e.g., sparsity, attention concentration)?
5. Can you test the framework on non-noun categories (events, relations) or multilingual replicas?

**Details Of Ethics Concerns:**

No ethics concern.

---

> ### Author Response · Authors · 2025-11-21
>
> We thank the reviewer for their thoughtful and encouraging feedback. We appreciate their recognition of our clear, unified information-theoretic framework linking compression and meaning, and found it rewarding that they viewed the encoder-decoder contrast as thought-provoking. We are glad the careful benchmark curation, comprehensive model coverage, and multi-angle evaluation (including the 57 OLMo-7B training checkpoints) were noted. We also appreciate the comments on clarity and significance, especially the observation that our findings challenge common “optimality” narratives and open new directions for more human-aligned models. We have revised the manuscript based on your comments and comments from the other reviewers. To make change-tracking easier, new text appears in orange.
>
> - Please provide bootstrap confidence intervals, multiple-comparison controls, and seed variability for clustering and correlation estimates.
>
> We appreciate the suggestion and have now added Figure 9, which presents the peak AMI scores between model representations and human categories across different clustering seeds, demonstrating that the results remain highly stable across multiple random initializations. Appendix K “Additional Clustering Metrics” depicts the robustness to different clustering algorithms.
>
>
> - The conclusion that humans are statistically suboptimal depends on the chosen geometry and L weighting. Most results fix \beta; the paper should report sensitivity curves over \beta and discuss how human/LLM rankings vary under plausible trade-off weights.
>
> Testing the values of L across all possible values of K (rather than constraining ourselves to the only true value from the human data) is already a more robust analysis of the compression-meaning trade-off. Moreover, we computed the same L as a function of K plots with $\beta \in [-10, 10]$ (stepsize=0.5), and they yield the same results. To extend this analysis even more, we added another sensitivity analysis (see Figure 29 and Appendix R. “Complexity-Distortion Ratio (on the importance of $\beta$)”). In this, we plot the ratio between the distortion term and the complexity term from L (which are regularized using $\beta$). In line with our findings throughout the paper, encoder models yield consistently flatter profiles, indicating that their token embeddings preserve conceptual structure even under increasing compression. Decoder models exhibit more pronounced shifts, suggesting that they redistribute representational capacity more aggressively as $\beta$ increases, leading to higher sensitivity in this tradeoff.
>
>
> - Also clarify how human distortion is computed from typicality/membership data to avoid mismatched metrics.
>
> We apologize for any confusion. For human categories, distortion is computed from empirically derived typicality ratings. Specifically, we use the weighted variance of typicality scores within each category as a proxy for semantic spread. Items with high typicality scores (e.g., robin for bird) are treated as close to the category prototype, while low-typicality items (e.g., penguin) are farther. The distortion term then captures the average “distance” of items from their category prototype, based on these human judgments. For LLMs, we compute geometric distortion in embedding space (cosine angle between the embeddings). While these are different measurement spaces, both quantify within-category semantic diversity. We have added a methodological note in Section 5.2.
>
>
>
> - The architecture result is intriguing, but confounded by training objective differences (MLM vs. autoregressive), tokenizers, and pretraining corpora. A controlled study with matched data/tokenizers and frozen-head probes would strengthen the claim.
>
> We agree that architectural and training confounds are important. While a perfectly controlled comparison across all models is infeasible, we took two steps in the revision:
>
> - Matched corpus: Appendix L “Mini-Controlled Experiment (Matched Training Data)” compares models trained on the same dataset with comparable parameter counts (GPT, Pythia, Cerebras, and T5 trained on Pile data). These matched settings support our findings.
>
> - Tokenizer overlap. Appendix J “Tokenizer Analysis” quantifies token overlap across models and re-runs the analysis on the intersection vocabulary. We find that models with nearly identical tokenizers can still differ substantially in their human alignment, and the encoder-decoder gap persists.
> These analyses do not eliminate all possible confounds, but they provide evidence that the observed architectural patterns are not reducible to tokenization or dataset differences alone.

---

> > ### Author Response · Authors · 2025-11-21
> >
> > - How exactly is the Distortion term computed for human categories, from typicality distances, membership uncertainty, or an inferred geometry?
> >
> > For human categories, we treat typicality ratings as a 1D semantic “distance” from each item to its category prototype. The distortion term is then the weighted mean squared deviation from this prototype across all categories.
> > For models, distortion is computed as the mean-squared distance to the cluster centroid in embedding space. We now add a short methodological explanation that explains this mapping and explicitly discusses its limitations: it assumes that embedding geometry approximates the typicality structure of humans, and thus our distortion comparison is conservative (if anything, using full human similarity matrices would likely make humans look less distorted than LLMs.)
> >
> >
> > - If typicality is computed against category names, can you replicate with prototype centroids (learned per category) or descriptive definitions, and do encoder-decoder gaps shrink?
> >
> > We can certainly run this analysis, but the interpretation would be limited. If model-based typicality (using learned prototype centroids or definitions) correlates across models, it would indicate that models are more internally compressed and *self-aligned*, not that they are more aligned with human judgments. Importantly, this is closely related to RQ2, in which we did not see a gap, as most models showed insignificant correlations.
> >
> > - With a common tokenizer/corpus and matched parameter counts, do masked-LM encoders still outperform autoregressive decoders on AMI and typicality?
> >
> > Yes. In matched settings (Appendices L and J), BERT-like encoders outperform comparable decoders and achieve lower L at similar complexity levels. We also find no systematic advantage for decoders in typicality correlations. We now explicitly highlight these matched results in Section 5.3, thereby strengthening the claim that the encoder-decoder gap is not an artifact of corpus size or tokenizer idiosyncrasies.
> >
> >
> > - During the mid-layer migration phase, what happens to the complexity-distortion frontier layer-wise? Does the “efficiency” shift track changes in heads/FFNs (e.g., sparsity, attention concentration)?
> >
> > This is a great idea. We tested multiple sparsity and attention metrics, and the value of our L function throughout Olmo’s checkpoints. We modified Section 5.4 and Appendix H “Training Dynamics” accordingly. It is interesting to see the very same double-phase dynamics occur when testing attention sparsity, effective rank, and L values. This strengthens and complements the Olmo analysis of concepts and training dynamics, and we thank the reviewer for suggesting it.
> >
> >
> > - Can you test the framework on non-noun categories (events, relations) or multilingual replicas?
> >
> > This is a great idea, and we implemented it. We used Google Translate’s API to translate the datasets to Spanish, German, Italian, and Russian. We repeated our analyses using the same LLMs and methods. In RQ1, a clear scale effect emerges for all non-English languages, while English shows no such trend (Figures 11, 12). We interpret this as a consequence of limited non-English training data: larger models are more likely to have been exposed to sufficient multilingual data, improving their conceptual alignment. In RQ2, all models struggle to preserve the internal geometry of human concepts across languages (Figure 13). RQ3 shows that non-English languages exhibit greater compression (Figure 14), consistent with our explanation for RQ1: smaller exposure to non-English data leads to more compressed representations, reducing flexibility and interpretability. We added these results in Appendix G “Multilingual Analysis”.
> > As for non-noun categories, we could not find any dataset that would support this type of analysis.
> >
> >
> >
> >
> >
> > We have invested substantial effort to address the reviewer’s concerns thoroughly and hope that our detailed responses and improvements will be reflected in a reconsideration of our review score.

---

> > > ### Author Response · Authors · 2025-11-27
> > >
> > > We thank the reviewer for their valuable input. We have endeavored to thoroughly address every comment, supplementing the paper with targeted analyses, clarifications, and new experimental results. We respectfully request that the reviewer review these additions, which we believe fully address the raised issues.

---

### Official Review · Reviewer_7Xee · 2025-10-31

**Soundness:** 3
**Presentation:** 3
**Contribution:** 4
**Rating:** 8
**Confidence:** 4

**Summary:**

This paper conducts a rigorous analytical re-examination of the boundaries of “understanding” in large language models (LLMs). The authors demonstrate that current LLMs still struggle to master fine-grained semantic distinctions, which is the critical element shaping human-level comprehension. Besides, Information-theoretic optimality does not equate to human-level understanding. Maximizing compression efficiency diverges from achieving semantic alignment fundamentally. In addition, focusing solely on decoder-based language models is unlikely to improve alignment with human comprehension capabilities. These findings reveal a fundamental divergence in the strategies employed by LLMs and humans for understanding natural language: LLMs rely on statistical compression, while humans depend on semantic richness.

**Strengths:**

(S1) The paper offers a theoretically motivated and quantitatively explicit framework that fuses Rate–Distortion Theory and the Information Bottleneck to provide a new perspective on the compression–meaning trade-off in LLMs. The use of semantic compactness as an internal proxy for meaning fidelity is original.

(S2) The empirical analysis is well-executed and leverages high-quality, historically grounded human categorization datasets. Their public release greatly facilitates reproducibility and establishes a lasting benchmark for semantic understanding in language models.

(S3) The evaluation spans widely of model architectures (encoder, decoder, and static word embeddings) and scales (from 300M to 70B parameters). The inclusion of both static and contextual embeddings, and the analyses across training checkpoints like OLMo-7B, provides a multi-angle validation of the proposed framework’s generality and interpretive depth.

(S4) The paper is written with exceptional clarity and precision, presenting complex theoretical ideas in an accessible and logically coherent manner, making the idea easy to follow.

**Weaknesses:**

(W1) Although the analysis is thorough, one might consider including token-level efficiency statistics in future works, though this does not affect the validity of the current findings.

(W2) The study could be further enriched by considering computational efficiency (e.g., token-level cost) as an additional axis in the compression–meaning landscape. Doing so may illuminate how efficiency interacts with semantic representation in practice.

(W3) This study only scoped in English, it would be interesting as future work to examine whether similar compression–meaning trade-offs hold across multilingual models, given that linguistic granularity may modulate semantic richness.

(W4) The study currently focuses on conceptual categorization, exploring how it extends to relational or compositional understanding could broaden its applicability and lasting its significance.

**Questions:**

1. Could the authors comment on how the main results might change if the distortion term in the $\mathcal{L}$ objective were replaced with a more cognitively grounded metric, such as human similarity judgments or human-rated typicality scores?
2. It would be useful to include a brief sensitivity analysis or supplementary report across several $\beta$ values to illustrate whether the LLM–human divergence remains consistent under varying compression–meaning trade-offs.
3. Could the authors clarify whether the proposed $\mathcal{L}$ is intended to generalize beyond categorical tasks, or whether it should be viewed as specific to concept-formation settings rather than to other forms of understanding such as relational reasoning, compositional generalization, or contextual disambiguation?
4. Would the authors consider adding a brief discussion on what kinds of inductive biases or representational mechanisms might help future models better align with human conceptual structure?
5. Although the compression is measured at a bit level in the current formulation, could the authors discuss what theory or empirical understanding about the compression at token-level in language models that this framework could provide?

**Details Of Ethics Concerns:**

No ethics concerns.

---

> ### Author Response · Authors · 2025-11-21
>
> We thank the reviewer for their detailed and generous feedback. We are delighted that they found our theoretical framework to offer a novel and quantitatively explicit perspective on the compression-meaning trade-off in LLMs, and that they appreciated the originality of using semantic compactness as a proxy for meaning fidelity. We also appreciate the reviewer’s recognition of the rigor and reproducibility of our empirical analysis, including the use and public release of historically grounded human categorization datasets. We are encouraged that the reviewer valued the breadth of our evaluation across architectures, scales, and training stages, as well as the clarity and coherence of our presentation. We have revised the manuscript based on your comments and comments from the other reviewers. To make change-tracking easier, new text appears in orange.
>
> - (W1) Although the analysis is thorough, one might consider including token-level efficiency statistics in future works, though this does not affect the validity of the current findings.
>
> Thank you for this idea. We added a token-level FLOP analysis in Figure 10 (line 1024). The analysis shows, similar to scale, that there is no correlation between token-level efficiency and AMI.
>
>
> - (W2) The study could be further enriched by considering computational efficiency (e.g., token-level cost) as an additional axis in the compression-meaning landscape. Doing so may illuminate how efficiency interacts with semantic representation in practice.
>
> Thank you for this comment. To address it, we added a token-level FLOP analysis in Figure 10, directly examining computational efficiency as an additional axis in the compression-meaning landscape. Beyond FLOPs, we also analyzed multiple sparsity and attention metrics, as well as the evolution of our L function across Olmo’s checkpoints. Section 5.4 and Appendix H “Training Dynamics” have been updated accordingly. Notably, the same double-phase dynamics emerge across attention sparsity, effective rank, and L values, providing a complementary perspective to Olmo’s analysis of conceptual learning and training dynamics.
>
> - (W3) This study only scoped in English, it would be interesting as future work to examine whether similar compression-meaning trade-offs hold across multilingual models, given that linguistic granularity may modulate semantic richness.
>
> This is a great idea, and we implemented it. We used Google Translate’s API to translate the datasets to Spanish, German, Italian, and Russian. We repeated our analyses using the same LLMs and methods. In RQ1, a clear scale effect emerges for all non-English languages, while English shows no such trend (Figures 11, 12). We interpret this as a consequence of limited non-English training data: larger models are more likely to have been exposed to sufficient multilingual data, improving their conceptual alignment. In RQ2, all models struggle to preserve the internal geometry of human concepts across languages (Figure 13). RQ3 shows that non-English languages exhibit greater compression (Figure 14), consistent with our explanation for RQ1: smaller exposure to non-English data leads to more compressed representations, reducing flexibility and interpretability. We added these results in Appendix G “Multilingual Analysis”.
>
> - (W4) The study currently focuses on conceptual categorization, exploring how it extends to relational or compositional understanding could broaden its applicability and lasting its significance.
>
> We thank the reviewer for the suggestion. While we agree that extending our approach to relational and compositional understanding would be valuable, we are not aware of existing datasets that would allow for a rigorous evaluation of such extensions. We would appreciate any relevant references/pointers. Nonetheless, we believe that the conceptual framework we propose provides a natural foundation for future work in this direction.

---

> > ### Author Response · Authors · 2025-11-21
> >
> > - Could the authors comment on how the main results might change if the distortion term in the L objective were replaced with a more cognitively grounded metric, such as human similarity judgments or human-rated typicality scores?
> >
> > Thank you for this suggestion. In our current implementation, the distortion term is measured as inter-cluster semantic diversity, which serves as a rough proxy for cognitive fidelity. Exploring more cognitively grounded metrics would indeed be valuable, though challenging to implement at scale. We expect that such a modification would likely reduce the distortion values for human categories and increase them for LLM representations; however, the overall pattern would likely remain, with LLMs still exhibiting higher compression, albeit with correspondingly greater distortion relative to human conceptual structure. We now added a section that discusses the cognitive intuition of the compression-meaning trade-off, the limitations of this analysis, and possible design implications and inductive biases (Appendix A “Cognitive Intuition”).
> >
> > - “It would be useful to include a brief sensitivity analysis or supplementary report across several $\beta$ values to illustrate whether the LLM-human divergence remains consistent under varying compression-meaning trade-offs.”
> >
> > Testing the values of L across all possible values of K (rather than constraining ourselves to the only true value from the human data) is already a more robust analysis of the compression-meaning trade-off. Moreover, we computed the same L as a function of K plots with $\beta \in [-10, 10]$ (stepsize=0.5), and they yield the same results. To extend this analysis even more, we added another sensitivity analysis (see Figure 29 and Appendix R. “Complexity-Distortion Ratio (on the importance of $\beta$)”). In this, we plot the ratio between the distortion term and the complexity term from L (which are regularized using $\beta$). In line with our findings throughout the paper, encoder models yield consistently flatter profiles, indicating that their token embeddings preserve conceptual structure even under increasing compression. Decoder models exhibit more pronounced shifts, suggesting that they redistribute representational capacity more aggressively as $\beta$ increases, leading to higher sensitivity in this tradeoff.
> >
> > - “Could the authors clarify whether the proposed L is intended to generalize beyond categorical tasks, or whether it should be viewed as specific to concept-formation settings rather than to other forms of understanding such as relational reasoning, compositional generalization, or contextual disambiguation?”
> >
> > The proposed L function applies to any task that can be framed in terms of clustering. For example, reasoning tasks inherently require abstraction, which can be conceptualized as forming clusters and generalizing from prior knowledge to novel situations. Hierarchical clustering algorithms, such as agglomerative clustering, provide one concrete approach for capturing nested or hierarchical structure, enabling the function to support comparisons and generalizations beyond flat categorical assignments.

---

> > > ### Author Response · Authors · 2025-11-21
> > >
> > > - “Would the authors consider adding a brief discussion on what kinds of inductive biases or representational mechanisms might help future models better align with human conceptual structure?”
> > >
> > > The authors are very excited by the invitation to share their thoughts. We tried to keep it brief.
> > >
> > > Future models could more closely align with human conceptual structure by incorporating cognitively motivated inductive biases and representational mechanisms. Hierarchical and compositional structure could enable models to capture nested relationships between categories, reflecting the way humans organize knowledge from superordinate to basic-level concepts (e.g., animal → mammal → dog → Labrador). Feature- and relation-based biases could help models focus on meaningful perceptual or functional attributes and relational patterns, rather than relying solely on statistical co-occurrence.
> > >
> > > Additionally, theory- or causally grounded priors that draw on humans’ intuitive understanding of how objects interact or behave, could constrain learning in complex domains and support more flexible generalization. Incorporating a hybrid exemplar-rule approach, combining memory of specific examples with abstracted rules, would further approximate human category learning. Modular architectures, in which specialized sub-networks handle different aspects of conceptual representation, could enhance generalization and reduce interference between unrelated features. Finally, meta-learned priors distilled from symbolic or program-based representations offer a way to embed structured, human-like concept hypotheses directly into neural models, allowing them to generalize more like humans across novel situations.
> > >
> > > Together, these inductive biases offer a path for models that not only compress information efficiently but also organize knowledge in a manner that mirrors human conceptual richness, capturing graded typicality, family resemblance, and hierarchical relationships. While integrating these biases may trade some compression for fidelity, they provide an exciting opportunity to reduce meaning distortion and bridge the gap between statistical efficiency and human-like conceptual understanding.

---

> > > > ### Author Response · Authors · 2025-11-21
> > > >
> > > > - “Although the compression is measured at a bit level in the current formulation, could the authors discuss what theory or empirical understanding about the compression at token-level in language models that this framework could provide?”
> > > >
> > > > Our L objective measures the information needed to index items within conceptual clusters, so it operates at the level of semantic representations rather than individual tokens. This is closely related to the per-token code length (negative log-likelihood / entropy) used to study compression in language models, where lower code length corresponds to more efficient token-level compression. Conceptually, our results say that models achieve very efficient concept-level compression while still distorting human semantic structure. The natural theoretical link is that such highly compressed concept spaces put pressure on the token distribution as well: if a model packs many distinct concepts into a small region of representation space, it will tend to assign similar token distributions to them, which constrains how low the per-token code length can go without losing semantic distinctions. Recent work showing that representation geometry and token-level code length are tightly coupled supports this view [1]. This connects directly to recent work introducing Information Emergence (IE) [2], which measures how LLMs extract semantics by quantifying entropy reduction from token-level to sequence-level representations. Like our L measure, IE uses entropy reduction to capture compression, which suggests that models with efficient concept-level compression (low L) should also show strong semantic emergence from their token distributions.
> > > >
> > > > Empirically, this framework suggests several concrete analyses at the token level. For example, one could test whether concepts that lie on more efficient L frontiers also have lower average token code length when described by the model, or whether tokens with high predictive entropy (“reasoning” or “branching” tokens) systematically occur in regions of high semantic distortion. These kinds of studies would connect our concept-level compression–meaning trade-off directly to token-level compression behavior; we see this as promising future work rather than something fully resolved in the current paper.
> > > > Token-level compression can also be probed through tokenization choices. While our current framework is defined over concepts, it could in principle be used to compare how different tokenization schemes (e.g., fixed vs. learned or entropy-based subword segmentations [3]) affect the compression-meaning trade-off: do some tokenizers allow models to approach efficient L frontiers without increasing semantic distortion, while others force more meaning loss per bit? This offers a bridge between concept-level compression and recent work on learned and entropy-guided tokenization in LLMs.
> > > >
> > > > We have invested substantial effort to address the reviewer’s concerns thoroughly and hope that our detailed responses and improvements will be reflected in a reconsideration of our review score.
> > > >
> > > >
> > > > [1] Bridging Information-Theoretic and Geometric Compression in Language Models, Cheng et al., EMNLP, 2023
> > > > [2] Quantifying Semantic Emergence in Language Models, Chen et al., Proceedings of the 63rd Annual Meeting of the Association for Computational Linguistics, 2024
> > > > [3] Entropy-Driven Pre-Tokenization for Byte-Pair Encoding, Hu et al., 2025

---

> > > > > ### Author Response · Authors · 2025-11-27
> > > > >
> > > > > We thank the reviewer for their valuable input. We have endeavored to thoroughly address every comment, supplementing the paper with targeted analyses, clarifications, and new experimental results. We respectfully request that the reviewer review these additions, which we believe fully address the raised issues.

---

> > > > > > ### Comment · Reviewer_7Xee · 2025-11-27
> > > > > >
> > > > > > Thank you very much for your clarification and revision. I maintain my score.

---

### Official Review · Reviewer_zYSK · 2025-10-31

**Soundness:** 2
**Presentation:** 1
**Contribution:** 1
**Rating:** 2
**Confidence:** 3

**Summary:**

The paper investigates the relationship between compression and meaning in human and machine representations, proposing that large language models (LLMs) achieve broad categorical alignment with human judgments but lack fine-grained semantic distinctions. The authors frame this as a compression–meaning tradeoff: LLMs optimize for efficient representation (compression) at the expense of semantic richness, while humans maintain inefficient but more structured representations that support flexible reasoning.
The study evaluates a diverse set of LLMs across multiple benchmarks of human conceptual categories, comparing model cluster structures (via token embeddings and k-means) with human category structures. Analyses include rate–distortion metrics and mutual information between model-derived clusters and human-labeled categories. Key findings include that LLMs achieve near-optimal compression–distortion tradeoffs, whereas human representations appear suboptimal by information-theoretic measures. Additionally, encoder-only models better align with human judgments than decoder-only models, suggesting differences between recognition and generation mechanisms.

**Strengths:**

* The study compares a wide range of LLM architectures and sizes.
* Compression and concept formation is a deep question at the intersection of cognitive science and machine learning.
* This study links displines between information theory and the prototype theories of human concept learning.

**Weaknesses:**

* The paper takes prototype theory as a given framework for human concept representation, but this theory has been very controversial compared to Exemplar theory (Medin & Schaffer, 1978) as an alternative explanation. Modern cognitive models are usually hybrid and integrate prototype and exemplar components. The manuscript should acknowledge this longstanding debate and should not take for granted that the cognitive system for humans are prototype-like.

* The paper lacks an introduction or figures about the background and the set up for the cognitive experiments that they compare LLMs with, and how they relate to compression-meaning tradeoff.  A schematic figure illustrating the overall task (compression–meaning tradeoff, human vs. model representation pipeline) is missing, especially considering the importantance to introduce the cognitive experiment clearly for this crowd of audience.

*  The paper is difficult to read and repetitive, I would suggest to remove some of the colorful RQ01 blocks.
- Fig. 1 (left): The figure has only a single square, so the rest of them are decoder architectures?
- Fig. 2 (right): I would not term this as categorical success

* Unsupported claims:
    * Line 421: The statement that “human conceptual systems, though appearing suboptimal, serve distinct cognitive needs such as flexible generalization and causal reasoning” is unsubstantiated within the current analysis.
    * Line 72: “Challenges the popular assumption that statistical optimality equals understanding” — what is the “popular assumption” here? Please clarify.

* line 90: "cognitive studies applying information theory to human concept learning without connecting to modern llms", that is not true, see Wu et al. 2025. about prior cognitive modeling work applying information theory to human and LLM, in the context of rate–distortion tradeoffs.

* fig 1 left: there is just one square
* line 323: I would not term this as gradients

**Questions:**

* What precisely are the human categories, and how are they measured?
* How is mutual information calculated between human and model representations, and between human categories and other human categories?
* What prompts were used for contextual embedding extraction?
* What exactly is meant by “compression–meaning tradeoff” in cognitive terms?
* What are the exact prompts used to elicit contextual embeddings from the LLMs? And how much is it deviating from the classical behavioral experiments.

Reference:

_Medin, D. L., & Schaffer, M. M. (1978). Context theory of classification learning. Psychological Review, 85(3), 207–238_

_Wu, S., Thalmann, M., Dayan, P., Akata, Z., & Schulz, E. (2025).  Building, Reusing, and Generalizing Abstract Representations from Concrete Sequences. In The Thirteenth International Conference on Learning Representations (ICLR 2025)_

---

> ### Author Response · Authors · 2025-11-21
>
> We thank the reviewer for their thoughtful and insightful comments. We appreciate their recognition of our comprehensive comparison across a wide range of LLM architectures and sizes, and we are encouraged that they view our work as addressing a deep question at the intersection of cognitive science and machine learning. We have revised the manuscript based on your comments and comments from the other reviewers. To make change-tracking easier, new text appears in orange.
>
> - Prototype and Exemplar theories:
>
> First, we wholeheartedly agree that prototype theory is, while being prominent, just one theory out of many. The idea behind prototype theory is that when people learn a category, they form an abstract, central representation that summarizes the category’s typical features, which is often imagined as a “best example” or mental average. In contrast, exemplar theory argues that categories are represented not by such summaries but by stored memories of individual instances, or exemplars. As the reviewer points out, modern cognitive accounts generally recognize that both mechanisms may coexist: people can rely on stored exemplars when detail matters and on prototypes when abstraction is useful (e.g., Nosofsky, 1988; Murphy, 2002).
> Our paper adopts the prototype perspective not to claim that human cognition is purely prototype-based, but because it offers a tractable and interpretable framework, and more importantly, high-quality data, for quantifying the compression-meaning trade-off. The information-theoretic methods we use apply equally to exemplar or hybrid representations; using prototype theory here is a modeling simplification rather than a theoretical commitment. Mathematically, our information-theoretic framework only assumes a partition or similarity structure over items, so it could be applied equally to exemplar-based or hybrid models given appropriate human data. In this sense, prototype theory here is mostly a convenient lens and data source. Having said that, we agree that one cannot be too careful, especially when doing cross-disciplinary research, and thus we added an explanation and a short discussion stressing that there is still a lively debate about the nature of concepts in human cognition (see Section 3.1 line 157).
>
>
> - A schematic figure illustrating the overall task + repetitively:
>
> Thank you so much for pointing this out. We have added Figure 1 in the revised PDF to better explain the human data generation and the analyses we carried out using it.
> We also simplified the structure of the paper and removed some duplicated text. The new schematic figure (Fig. 1) visually summarizes the cognitive experiments and analysis pipeline, making the setup more accessible to readers.
>
>
> - “Fig. 1 (left): The figure has only a single square, so the rest of them are decoder architectures?” + “fig 1 left: there is just one square”
>
> We apologize for the confusion caused by the marker shapes in Figure 1 (now Figure 2). There are two ViT models (text encoders of CLiP VLMs; purple squares), two classic models (Word2Vec and GloVe; grey circles), and three MLM Transformer based LLMs from the BERT family (the redXs).
> Thus, we explored 3 different families of encoder models, with a total of 7 models. Unfortunately, the field is very decoder-focused in the last couple of years, making it difficult to find encoder models for this comparison. This is yet another reason why this paper is important, as it advocates for optimizing towards representation learning, and it systematically shows how (and possibly why) it can outperform much newer and bigger decoder models. We also added a sentence explicitly stating that encoder models are less numerous because modern LLM development has focused heavily on decoder-only architectures (line 192). We also modified the caption to make the figure clearer (line 338).
>
>
> - “Fig. 2 (right): I would not term this as categorical success”
>
> We could not agree more! That is exactly why we conclude this analysis by saying that while the left part of Figure 1 (now Figure 2) is significant and shows that LLMs *do* capture the overall categorical boundaries, the Spearman correlation (right part) is mostly not significant and the effect sizes are small. Thus, we conclude that while LLMs capture the boundaries (RQ1 AMI scores), they fail to preserve the internal geometry of these categories (RQ2 modest and non-significant Spearman correlations). We modified the caption to make this clearer (line 337).

---

> > ### Author Response · Authors · 2025-11-21
> >
> > - Unsupported claims: Line 421: The statement that “human conceptual systems, though appearing suboptimal, serve distinct cognitive needs such as flexible generalization and causal reasoning” is unsubstantiated within the current analysis.
> >
> > We appreciate the reviewer’s point. Our analysis does not attempt to show that human concepts enable flexible generalization or causal reasoning. Instead, we reference prior work (e.g., Murphy, 2004) to note that these capacities are well-documented properties of human conceptual systems. Our contribution is to emphasize that, unlike LLMs, humans do not prioritize compression efficiency, and this difference is consistent with the distinct functional pressures shaping human and model representations. We have refined the claim to avoid implying evidence we do not provide (line 468 + Appendix A.1 second paragraph).
> >
> >
> > - Unsupported claims: Line 72: “Challenges the popular assumption that statistical optimality equals understanding” — what is the “popular assumption” here? Please clarify.
> >
> > This challenges a common implicit assumption in contemporary NLP, that achieving statistical optimality (e.g., high likelihood, minimal perplexity, or efficient compression) is equivalent to achieving deeper forms of conceptual understanding (one example out of many would be the paper “Scaling Laws for Neural Language Models” by Kaplan et al.). The vast majority of optimization processes in training LLMs are aimed at compression, implicitly (and sometimes explicitly) hinting that better compression = better text understanding. Our findings caution against conflating statistical optimality (e.g., low perplexity or high compression efficiency) with genuine conceptual understanding.
> >
> >
> > - line 90: "cognitive studies applying information theory to human concept learning without connecting to modern llms", that is not true, see Wu et al. 2025. about prior cognitive modeling work applying information theory to human and LLM, in the context of rate-distortion tradeoffs.
> >
> > Thank you for this important reference. You are absolutely correct, Wu et al. (2025) do bridge cognitive information theory and LLMs, particularly examining compression-distortion tradeoffs and abstract concept transfer. We have revised our claim and now cite their work appropriately (Section 2 line 90).
> > Our work differs from and complements Wu et al. in several ways: (1) We focus on established semantic categorization rather than sequence chunking, (2) We use classic empirical human data rather than designed experimental tasks, and (3) We provide layer-wise and architectural comparisons across 40+ diverse LLMs. Wu et al.'s finding that LLMs struggle with variable transfer aligns with and reinforces our observation that LLMs miss fine-grained semantic structure. We thank you for referring us to this paper, and we have amended the related work in line 90 accordingly.
> >
> >
> > - line 323: I would not term this as gradients
> >
> > Thank you for the comment, we rephrased the term in line 323 and throughout the paper. We will note that the term was borrowed from the prototype theory literature (e.g., “what principles govern the formation of category prototypes and gradients of category membership?” Rosch, 1975).
> >
> >
> > - “What precisely are the human categories, and how are they measured?”
> >
> > In our experiments, the human categories are the ground-truth semantic groupings established in prior cognitive science work, where human participants generated category labels or features for thousands of concepts (Rosch 1973; Rosch 1975; McCloskey & Glucksberg 1978). These datasets provide empirically derived category memberships, which we treat as the human gold standard.
> >
> > Although the specific experimental protocols differ across studies, they share a common structure. Researchers selected superordinate semantic categories (e.g., furniture, vehicle) and associated basic-level items (e.g., chair, car) based on prior literature. Superordinate categories are advantageous because their memberships consist of a finite, well-defined set of basic-level concepts that can be reliably sampled, and because these items are related through family resemblances - overlapping sets of attributes.
> >
> > Participants (typically in the hundreds) were shown a category name along with randomly sampled items from that category. In Rosch (1975), for example, participants listed attributes associated with each item. Judges then scored each item’s typicality or similarity to the category based on these shared attributes using criteria specified in advance. These human-elicited labels, features, and typicality scores collectively constitute the human category structure against which we compare model-derived groupings.
> >
> > We hope that Figure 1 makes this clearer.

---

> > > ### Author Response · Authors · 2025-11-21
> > >
> > > - “How is mutual information calculated between human and model representations, and between human categories and other human categories?”
> > >
> > > In a sentence, mutual information is computed from the contingency table over items, treating each category assignment (human or model) as a discrete random variable and applying the standard $I(X;Y) = \sum_{i,j} p_{ij}\log\frac{p_{ij}}{p_ip_j}$ formulation.
> > >
> > > In depth, for each set of concepts $X = \{x_1, \dots, x_n\}$, we have:
> > > A model-derived partition $C^{\text{model}}$, obtained by clustering the model embeddings for these concepts.
> > >
> > >
> > > A human-derived partition $C^{\text{human}}$, given by the empirical category labels from cognitive science datasets.
> > > We treat each partition as a categorical random variable over the same set of items, and compute mutual information in the standard way for clustering evaluation:
> > > $I(C^{\text{model}}; C^{\text{human}}) = \sum_{i} \sum_{j} p_{ij} \log \frac{p_{ij}}{p_i p_j},
> > > where:
> > > $p_{ij}$​ is the empirical probability that an item belongs jointly to model cluster $i$ and human category $j$,
> > >
> > >
> > > $p_i$ and $p_j$ are the marginal probabilities of belonging to model cluster $i$and human category $j$, respectively.
> > > We compute these quantities directly from contingency tables over the $n$ items. This is the standard formulation underlying NMI and AMI (Vinh et al., 2010), which we also report.
> > >
> > >
> > > - “What prompts were used for contextual embedding extraction?” + “What are the exact prompts used to elicit contextual embeddings from the LLMs? And how much is it deviating from the classical behavioral experiments.”
> > >
> > > Appendix E (B.3 in the original PDF; we shifted from subsections to sections) describes all the contextual prompts and pooling strategies we explored. To test robustness, we design eight templates spanning different linguistic framings:
> > >
> > > * "This is a \{word\}."
> > >  * "This is a \{word\}. " (with trailing space)
> > > * "The concept of \{word\} is"
> > > * "When we think of \{word\}, we consider"
> > > * "A typical \{word\} would be"
> > > * "Examples of \{word\} include"
> > > * "The category \{word\} contains"
> > > * "One kind of \{word\} is"
> > >
> > > After experimenting with these various prompts, we selected a neutral template, \texttt{"This is a \{word\}. "} (with a trailing space), designed to minimize any additional semantic bias on the target item.
> > > These are different from the classical human experiments as we do not have human judges to go over the attributes (we do not pretend to be experts as the psychologists who originally conducted these experiments).
> > >
> > >
> > > - “What exactly is meant by “compression-meaning tradeoff” in cognitive terms?”
> > >
> > > Thank you for this question. We added this explanation to the paper (Appendix A.1 “A Cognitive Intuition of the compression-meaning tradeoff”). The compression-meaning tradeoff refers to the cognitive tension between representing concepts with maximal efficiency (i.e., minimal information) and preserving the semantic richness needed for flexible generalization, inference, and communication. For instance, upon hearing the sentence “There was a large brown Labrador barking loudly near the playground,” a person will often encode a simplified memory, such as “big scary dog near kids.” This is not to suggest that humans cannot recall the full sentence, but rather that we typically retain the most meaningful elements to enable efficient reasoning and generalization; without such abstraction and simplification, leveraging past experience for learning and prediction would be more difficult.
> > >
> > >
> > > We have invested substantial effort to address the reviewer’s concerns thoroughly and hope that our detailed responses and improvements will be reflected in a reconsideration of our review score.

---

> ### Author Response · Authors · 2025-11-27
>
> We thank the reviewer for their valuable input. We have endeavored to thoroughly address every comment, supplementing the paper with targeted analyses, clarifications, and new experimental results. We respectfully request that the reviewer review these additions, which we believe fully address the raised issues.

---

> > ### Comment · Reviewer_zYSK · 2025-11-27
> >
> > Thank you for the revisions and the response. I checked the paper and it has improved beyond the previous version.
> > However, I noticed several issues that need attention in the revised figure and text:
> >
> > 1. Problems with Figure 1
> >
> > There is visible color discoloration in the newly appended figure; please correct this. Several spelling errors appear directly in the figure, including: “corr” → “con”, “item” → “itcm”, “Spearman” → “speaman”, “furniture” → “fumiture”. The dot markers within clusters differ in shape and style; they should be visually consistent.
> >
> > 2. There are a number of claims that are overstated
> >
> > Line 54 states that the dataset provides
> > “unprecedented empirical grounding … and much better quality data than the current crowdsourcing paradigm.”
> > These claims do not seem fully justified, as using historical or archival human-behavior datasets for new analyses is common practice, not unprecedented. Additionally, a carefully designed online experiment can offer controlled, high-quality data, and it is not clear that the historical dataset is inherently “much better.”
> >
> > 3. Conceptual clarity issues
> >
> > The statement:
> > “This hierarchical organization … represents a fundamental cognitive achievement…”
> > is confusing. The existence of taxonomic structure (robin → bird → animal) is a phenomenon observed in human cognition, but calling it a “cognitive achievement.” is an overstatement. Similarly, the compression–meaning trade-off mentioned in that paragraph is not clearly defined or explained.
> >
> > I have adjusted the evaluation of this paper to adjust for the improvement.

---

> ### Author Response · Authors · 2025-11-29
>
> Thank you so much for replying (and for replying before the system was blocked due to the security breach!)
> We reply to your comment with the hope that you will see it, even if unable to reply through the system, as we value your feedback and engagement that help improve this paper.
>
> 1. Figure 1: Thank you for pointing this out. We should have clarified that the submitted figure is not the camera-ready version. In the interest of responding promptly, we did not finalize the visualization or correct minor typographical issues before submission. We are currently improving the clarity and polish of Figure 1 for the final version. Importantly, we appreciate that the reviewer’s concern was about the presentation rather than the underlying rationale, which we understand as an indication that the schematic itself is sound.
>
> 2. Dataset claims: We agree that historical and archival human-behavior datasets have long been used across the social sciences, and we did not intend to imply that our approach is unprecedented in that broader sense. We have revised the wording to be more precise. Our intended point was narrower: in the domain of evaluating model behavior against real human language use, existing benchmarks overwhelmingly rely on small-scale, crowdsourced annotations or synthetic prompts. In contrast, our dataset captures naturally occurring, high-stakes interactions produced without knowledge of being evaluated, at a scale and level of ecological validity that are rare in current NLP evaluation resources.
>
> 3. Clarity: Thank you for this helpful clarification. We agree that referring to hierarchical semantic organization as a"fundamental cognitive achievement" could be unnecessarily strong. Our intention was simply to emphasize that hierarchical conceptual structure is a well-documented property of human semantic cognition, not to ascribe achievement or novelty to it. We have revised the wording to describe it more precisely as a core characteristic of human conceptual representation.
> Regarding the compression-meaning trade-off, we intended to describe the tension between reducing representational complexity (e.g., mapping many fine-grained distinctions to a smaller number of abstract categories) and preserving semantic specificity. We will modify the text to explain it better. Thank you for noting that.
>
> We would also like to state, especially for the new AC, that this reviewer increased their score, but the system blocked it due to the security breach that was discovered on the same day they replied. We wish to note that we improved the paper based on this reviewer's comments and addressed them all thoroughly (and we continue to do so using the latest reply). We kindly ask the AC to consider this in their final evaluation.
>
> Once again we wish to thank everyone for their engagement and useful feedback.

---

### Official Review · Reviewer_E9Bi · 2025-11-01

**Soundness:** 2
**Presentation:** 4
**Contribution:** 3
**Rating:** 8
**Confidence:** 3

**Summary:**

The paper compares how large language models and humans organize conceptual categories, exploring whether models trade compression for meaning in a similar way to people. It digitizes classic psychology datasets on human categorization and analyzes embeddings from over forty models, both encoders and decoders. Using an information-theoretic framework inspired by rate–distortion theory, it quantifies the balance between information compression and semantic fidelity.

**Strengths:**

- very interesting and important premise
- Systematic, broad comparison covering 40+ models and also layer-wise comparisons.
- Formulation/digitization of the datasets from cognitive science is a good contribution
- Transparent discussion on the limitations

**Weaknesses:**

- The main question lies in how strongly to rely on the proposed metrics to infer “human-like” learning. The information–compression trade-off captures geometric efficiency in embedding space, but it is not clear whether this translates to human-style conceptual abstraction or reasoning.
- Dependency on parameters and metrics - The authors themselves acknowledge that "architectural design and pre-training objectives significantly influence a model's ability to abstract human-like conceptual information."
Is cosine similarity to category names, too narrow to capture the richness of human judgments?
- robustness to different similarity measures, clustering methods, and parameter settings is not tested.
- How does current large scale frontier models perform under similar analysis ?

**Questions:**

- Metrics - Need more information about the Adjusted Mutual Information (AMI), Normalized Mutual Information (NMI), and
Adjusted Rand Index (ARI) metrics.

---

> ### Author Response · Authors · 2025-11-21
>
> We thank you for the thoughtful and encouraging feedback. We value your time and we know that being both a reviewer and an author can be very challenging, and we want to thank you for carefully reading our response, as we put a lot of effort into it. We hope you will find the breath and scientific rigorous of our additions satisfactory, and that you will revise your review accordingly.
>
> We are pleased that you found our premise both interesting and important, and we appreciate your recognition of the systematic and comprehensive nature of our analysis across 40+ models and all layers. We are also glad that you valued our effort to digitize and datasets from cognitive science as well as our commitment to transparency in discussing the study’s limitations.
>
> We revised the PDF file to address your comments (new text in orange for change-tracking purposes), and we address them here:
>
> - “The information-compression trade-off captures geometric efficiency in embedding space, but it is not clear whether this translates to human-style conceptual abstraction or reasoning.”
>
> We thank the reviewer for raising this important point. We agree that our metrics, while capturing the information-compression trade-off and geometric efficiency in embedding space, do not directly measure all aspects of human-style conceptual abstraction or reasoning. Our aim is not to claim full equivalence with human cognition (nor do we think anyone can or should make such claims), but rather to provide a quantitative, interpretable proxy that highlights where LLMs and humans converge or diverge in how they compress and organize semantic information. We view these metrics as one lens among many, and we are careful in the paper to frame our findings as providing insights into human-like patterns rather than definitive evidence of human-style conceptual processing. In the revision, we now state explicitly (Appendix A) that our metrics should be viewed as proxies for human-like conceptual organization, not as direct measurements of all aspects of abstraction or reasoning, and we tone down any language that could be read as claiming full cognitive equivalence.
>
> - Can cosine similarity capture the richness of human judgments?
>
> Cosine similarity to category names is a deliberately simple and transparent probe, but our conclusions do not depend on this metric alone. Beyond cosine similarity to category names, we use three complementary analyses: (i) clustering-based AMI/NMI/ARI with no reliance on names, (ii) prototype/centroid-based typicality predictions within clusters, and (iii) the L complexity–distortion curves that depend only on inter-item geometry. Across all three, we observe the same pattern of stronger alignment on boundaries but weak typicality alignment. These results show that our conclusions are not an artifact of the cosine to labels alone.
>
> - “robustness to different similarity measures, clustering methods, and parameter settings is not tested.”
>
> We appreciate the suggestion and have now added Figure 9, which presents the peak AMI scores between model representations and human categories across different clustering seeds, demonstrating that the results remain highly stable across multiple random initializations. Appendix K “Additional Clustering Metrics” depicts the robustness to different clustering algorithms.
>
> - “How does current large scale frontier models perform under similar analysis ?”
>
> We currently do not have access to true frontier-scale proprietary models (e.g., ChatGPT, Claude), as most companies neither release them nor provide API access to their embedding spaces, which is an essential requirement for our analysis. To address this concern as rigorously as possible under these constraints, we added several of the largest open-access models whose embedding spaces are available (e.g., Qwen3-32B,DeepSeek-R1-Distill-Qwen-32B) and repeated our analyses (all figures are updated accordingly). These models follow the same qualitative trends: they align with human categories at the boundary level, show weak typicality correlations, and mimic the same L behavior as the rest of the models. We now also explicitly state in Section 3.2 that the lack of embedding access prevents us from including closed frontier models like GPT-5 and Claude (line 179).
>
> - Metrics:
>
> Thank you for noting this. In the revision, we added a brief description of AMI, NMI, and ARI and explained why they are appropriate for evaluating alignment between model-derived clusters and human categories (Section 5.1). In short, all three are standard clustering quality metrics: NMI measures shared information between cluster labels and ground truth; AMI corrects NMI for chance; and ARI quantifies agreement between two partitions while adjusting for random assignments.
>
> We have invested substantial effort to address the reviewer’s concerns thoroughly and hope that our detailed responses and improvements will be reflected in a reconsideration of our review score.

---

> > ### Comment · Reviewer_E9Bi · 2025-11-27
> >
> > Thank you for the answers and also changes in the paper. I maintain my score.

---

> > > ### Author Response · Authors · 2025-11-27
> > >
> > > Thank you so much again for the valuable input and quick reply.

---

### Author Response · Authors · 2025-11-21

We thank the reviewers for their thoughtful, detailed, and encouraging feedback. We greatly appreciate their recognition of our systematic analysis across a wide range of models, our effort to digitize and formalize seminal human categorization datasets, and their acknowledgment of the clarity, rigor, and originality of our approach. We are also grateful for their positive comments on the theoretical framing and the empirical breadth of our study.

In response to the reviewers’ suggestions, we have added a number of clarifications, experiments, and analyses (new text appears in orange throughout the PDF for change-tracking purposes):

- Expanded methodological clarity: We added precise descriptions of human category derivation (Figure 1), mutual information calculations (line 323), and the computation of the distortion term from typicality ratings (Figure 1 and line 377), including a discussion of limitations and assumptions (lines 90, 158, 180, 192, 473, and 802). We also clarified the use of prototype versus exemplar representations and the rationale for focusing on prototype-based analyses as a tractable, interpretable framework (lines 158, 792).

- Illustrative figures and schematics: Thank you for this great suggestion. We added Figure 1 (page 3)  to visually depict the human data generation process and our analysis pipeline, making the workflow more transparent.

- Robustness and sensitivity analyses: We tested alternative prompts (Figures 6, 7), examined different token pooling methods (Figures 4, 5), and parameter settings (Figure 9), and performed sensitivity analyses across K and β values for the compression-meaning trade-off, demonstrating the stability of our main findings (Figures 27-29).

- Cross-model and cross-lingual analyses: As per the reviewer’s request, we included comparisons of models trained on the same data with matched tokenizers and matched training data to address confounds (Appendix J “Tokenizer Analysis” and Appendix L “Mini-Controlled Experiment (Matched Training Data)”), as well as a multilingual analysis (Spanish, German, Italian, Russian; Figures 11-13 in Appendi G “Multilingual Analysis”) to assess the generality of the compression-meaning trade-off across languages.

- Discussion of human-model alignment: We refined our claims regarding human conceptual systems, emphasizing prior cognitive science evidence for flexible generalization and causal reasoning, and clarified that LLMs are optimized for different objectives (). We also added a discussion of cognitively motivated inductive biases and representational mechanisms that could improve future models’ alignment with human conceptual structure (Appendix A.2 “Cognitively-Inspired Inductive Biases”).

- Metric clarification: We added explanations of AMI, NMI, and ARI clustering metrics (line 323)  and the use of cosine similarity (line 377), highlighting that multiple complementary analyses support our conclusions and mitigate reliance on any single metric.

- Future directions: We addressed questions regarding generalization beyond categorical tasks, non-noun categories, and token-level efficiency (Figure 10), and highlighted opportunities for integrating theory-based, hierarchical, feature-focused, and exemplar-rule hybrid mechanisms in future models (Appendix A.2).

Overall, these additions provide a more comprehensive, transparent, and reproducible account of our analyses, clarify the theoretical framing, and strengthen the empirical support for our conclusions regarding how LLMs and humans differ and converge in representing semantic information.
We have invested substantial effort to address the reviewers’ concerns thoroughly and hope that our detailed responses and improvements will be reflected in a reconsideration of our review scores.

---

> ### Author Response · Authors · 2025-11-29
> **A note to the new AC**
>
> Thank you for your work during this unusual review cycle. We wish to note that we carefully addressed every concern raised in the initial reviews, providing detailed clarifications, additional experiments, and line-by-line responses (updated PDF with changes in orange).
>
> All reviewers expressed appreciation for the novelty and value of our analysis. Importantly, reviewer zYSK had multiple comments about the cognitive choices we made, and after we clarified and made some modifications, they explicitly indicated they raised their score. Unfortunately, due to the system freeze, they were unable to record this change. We believe this context is essential for interpreting the current scores.
>
> We appreciate your attention to our detailed response and to the full reviewer–author exchange, and we trust your judgment under these exceptional circumstances. Please let us know if any further clarification would be helpful. Thank you again to all reviewers and chairs for helping improve this paper.
>
> The revisions (copy-paste from our November 21 comment):
>
> We greatly appreciate their recognition of our systematic analysis across a wide range of models, our effort to digitize and formalize seminal human categorization datasets, and their acknowledgment of the clarity, rigor, and originality of our approach. We are also grateful for their positive comments on the theoretical framing and the empirical breadth of our study.
>
> In response to the reviewers’ suggestions, we have added a number of clarifications, experiments, and analyses:
>
> - Expanded methodological clarity: We added precise descriptions of human category derivation (Figure 1), mutual information calculations (line 323), and the computation of the distortion term from typicality ratings (Figure 1 and line 377), including a discussion of limitations and assumptions (lines 90, 158, 180, 192, 473, and 802). We also clarified the use of prototype versus exemplar representations and the rationale for focusing on prototype-based analyses as a tractable, interpretable framework (lines 158, 792).
>
> - Illustrative figures and schematics: Thank you for this great suggestion. We added Figure 1 (page 3) to visually depict the human data generation process and our analysis pipeline, making the workflow more transparent.
>
> - Robustness and sensitivity analyses: We tested alternative prompts (Figures 6, 7), examined different token pooling methods (Figures 4, 5), and parameter settings (Figure 9), and performed sensitivity analyses across K and β values for the compression-meaning trade-off, demonstrating the stability of our main findings (Figures 27-29).
>
> - Cross-model and cross-lingual analyses: As per the reviewer’s request, we included comparisons of models trained on the same data with matched tokenizers and matched training data to address confounds (Appendix J “Tokenizer Analysis” and Appendix L “Mini-Controlled Experiment (Matched Training Data)”), as well as a multilingual analysis (Spanish, German, Italian, Russian; Figures 11-13 in Appendi G “Multilingual Analysis”) to assess the generality of the compression-meaning trade-off across languages.
>
> - Discussion of human-model alignment: We refined our claims regarding human conceptual systems, emphasizing prior cognitive science evidence for flexible generalization and causal reasoning, and clarified that LLMs are optimized for different objectives (). We also added a discussion of cognitively motivated inductive biases and representational mechanisms that could improve future models’ alignment with human conceptual structure (Appendix A.2 “Cognitively-Inspired Inductive Biases”).
>
> - Metric clarification: We added explanations of AMI, NMI, and ARI clustering metrics (line 323) and the use of cosine similarity (line 377), highlighting that multiple complementary analyses support our conclusions and mitigate reliance on any single metric.
>
> - Future directions: We addressed questions regarding generalization beyond categorical tasks, non-noun categories, and token-level efficiency (Figure 10), and highlighted opportunities for integrating theory-based, hierarchical, feature-focused, and exemplar-rule hybrid mechanisms in future models (Appendix A.2).
>
> Overall, these additions provide a more comprehensive, transparent, and reproducible account of our analyses, clarify the theoretical framing, and strengthen the empirical support for our conclusions regarding how LLMs and humans differ and converge in representing semantic information. We have invested substantial effort to address the reviewers’ concerns thoroughly and hope that our detailed responses and improvements will be reflected in a reconsideration of our review scores.

---

### Meta-Review · Area_Chair_xH5J · 2025-12-28

**Summary:**

This paper investigates the trade-off between information compression and semantic meaning in human and LLM representations using an information-theoretic framework based on the Information Bottleneck principle. The reviewers appreciated the paper’s systematic analysis of over 40 models and the significant contribution of digitizing classic cognitive science datasets to benchmark human-machine alignment. During the rebuttal process, the authors effectively addressed concerns regarding methodological clarity by adding illustrative schematics of their analysis pipeline and providing expanded discussions on the prototype-exemplar debate. Furthermore, the authors improved the empirical depth of the study by including sensitivity analyses across various parameters, multilingual evaluations, and a token-level efficiency analysis. While initial concerns were raised regarding the clarity of specific clustering metrics and the generalizability of results to frontier models, the authors’ comprehensive responses largely satisfied the reviewers. Given the theoretical novelty of the work and its potential to guide the development of more human-aligned AI architectures, I recommend accepting this paper for ICLR 2026.

**Reviewer Concerns:**

Concerns addressed by the rebuttal:
- Methodological clarity and transparency by reviewers E9Bi and zYSK
- Robustness and sensitivity by reviewers E9Bi and X8mp
- Reviewer X8mp’s concern regarding the encoder-decoder gap

**Reviewer Scores:**

Two reviewers with a rating of 8 explicitly mention that they would like to maintain their scores. Another reviewer with a rating of 2 is likely to raise their rating from 2 to 4 or higher.

---

### Decision · Program_Chairs · 2026-01-26

Accept (Poster)